# Language Bias in LVLMs: From In-Depth Analysis to Simple and Effective Mitigation

**Yangneng Chen** [1]  **Jing Li** ✉ [1]

## Abstract

Large Vision-Language Models (LVLMs) extend large language models with visual understanding, but remain vulnerable to hallucination, where outputs are fluent yet inconsistent with images. Recent studies link this issue to *language bias*—the tendency of LVLMs to over-rely on text while neglecting visual inputs. Yet most analyses remain empirical without uncovering its underlying cause. In this paper, we provide a systematic study of language bias and identify its root in modality misalignment during training. Our analysis shows that both Visual Instruction Tuning (VIT) and Direct Preference Optimization (DPO) often prioritize textual improvements, which may cause LVLMs to overly lean toward language modeling rather than balanced multimodal understanding. To address this, we propose two simple yet effective methods: **Language Bias Regularization (LBR)**, which mitigates language bias through regularization during instruction tuning, and **Language Bias Penalty (LBP)**, which penalizes language bias in the DPO training process. Extensive experiments across diverse models and benchmarks demonstrate the effectiveness of our approach. LBR consistently improves performance on over ten general benchmarks, while LBP significantly reduces hallucination and improves trustworthiness. Together, these methods not only mitigate language bias but also advance the overall alignment of LVLMs, all without introducing any additional data or auxiliary models. Our code is publicly available at https://github.com/lab-klc/LVLM-Language-Bias.

[1]Harbin Institute of Technology, Shenzhen, China. Correspondence to: Jing Li <jingli.phd@hotmail.com>.

*Proceedings of the 43rd International Conference on Machine Learning*, Seoul, South Korea. PMLR 306, 2026. Copyright 2026 by the author(s).

**Vanilla Multimodal Training Process**

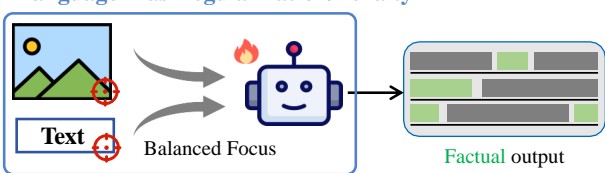

**Language Bias Regularization/Penalty**

*Figure 1.* Existing multimodal training paradigms (e.g., Visual Instruction Tuning, Direct Preference Tuning) often exhibit an over-reliance on text, which leads to language bias. To counter this, we propose two distinct methods, **Language Bias Regularization** and **Language Bias Penalty**, which encourage the LVLM to balance its focus between visual and textual modalities during training.

## 1. Introduction

The integration of vision into large language models (LLMs) has given rise to Large Vision-Language Models (LVLMs) (Liu et al., 2023a; 2024c), marking a pivotal step forward in multimodal artificial intelligence. However, this significant leap is shadowed by a persistent and critical challenge: hallucination (Sun et al., 2024; Zhou et al., 2024; Huang et al., 2024a; Bai et al., 2024). This failure mode, characterized by the generation of text that contradicts the visual input, fundamentally compromises the factual grounding of these models. Such unfaithfulness to the visual context not only degrades the reliability of LVLMs but also poses a significant barrier to their deployment in high-stakes, real-world applications.

Most studies (Wang et al., 2024; Yang et al., 2025; Yu et al., 2024c; Zhang et al., 2024), attribute LVLM hallucinations to a dominant *language bias*, where the model prioritizes linguistic fluency over visual consistency. A key manifestation of this bias, as illustrated in the top panel of Figure 2, is that the model allocates minimal attention to the image when generating lengthy responses. This indicates that the model, in essence, disregards the visual input, relying pre-

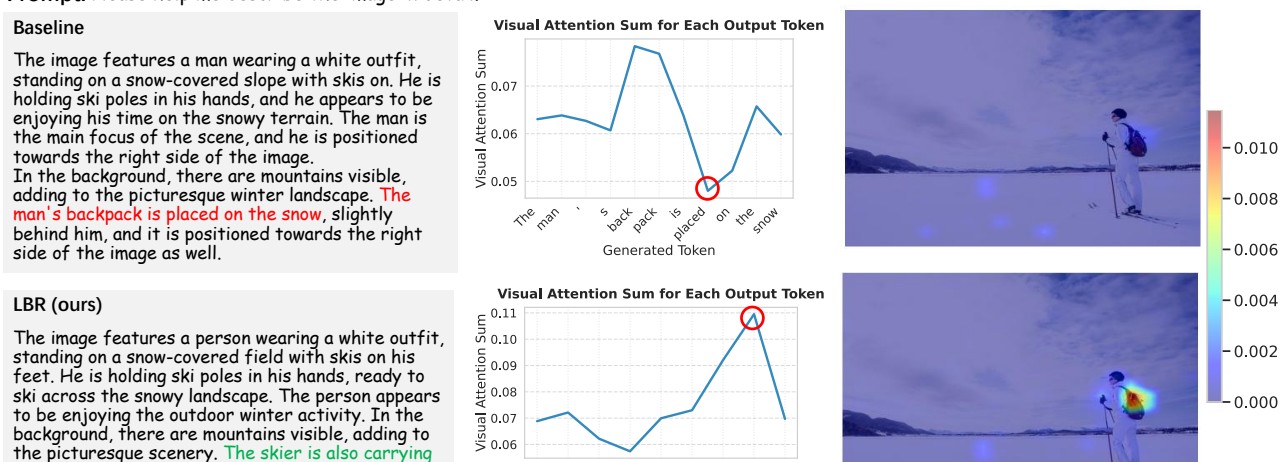

*Figure 2.* Demonstration of LBR's effect on mitigating language bias and preventing hallucinations. **(Top)** The baseline model shows low visual attention, especially on tokens that require strong visual grounding, resulting in hallucinations. **(Bottom)** Our LBR method enables the model to allocate substantially higher attention to these critical visual details, ensuring a factually accurate response.

dominantly on its internal language model. To counteract this, mitigation methods are typically categorized as either training-free, which focus on post-processing outputs (Chen et al., 2024b; Leng et al., 2024), or training-based, which address the issue during fine-tuning (Yu et al., 2024a;c; Yang et al., 2025).

Yet, the current understanding of language bias remains superficial and lacks a systematic, principled analysis. We address this gap by investigating its root cause, which we identify as a fundamental misalignment between the linguistic and visual modalities. Resolving this misalignment is crucial for both the trustworthiness and overall performance of LVLMs. Consequently, rather than focusing solely on hallucination scenarios, our work adopts a broader perspective to examine the underlying mechanisms driving this bias.

We begin by analyzing visual instruction tuning, a pivotal stage in aligning LVLMs. The central objective of this process is to maximize the conditional probability $\pi(y|x, v)$, where the model is expected to generate a response $y$ conditioned on both the instruction $x$ and the visual input $v$. However, our findings indicate that this paradigm places insufficient emphasis on the visual modality. In practice, models tend to over-rely on textual information, such that the improvement in the text-only likelihood $\pi(y|x)$ rivals—or even exceeds—that of the multimodal likelihood $\pi(y|x, v)$. A comparable tendency is also observed in Direct Preference Optimization (DPO) training. To capture this phenomenon, we formalize the **language bias** acquired during vision-language alignment as $\Delta\pi(y|x)$. Intuitively, this imbalance provides a principled explanation for the emergence of *language bias*: LVLMs systematically underutilize visual

signals, drifting toward behavior that resembles conventional language modeling, as illustrated in the top panel of Figure 1.

Building upon this insight, we explore loss function designs specifically targeting language bias. For visual instruction tuning, we propose **Language Bias Regularization (LBR)**, a simple yet effective term that encourages the model to focus more on vision-language alignment, thereby improving overall performance. For DPO training, we introduce the **Language Bias Penalty (LBP)**, which discourages the model from increasing its reliance on language-only cues and enhances its trustworthiness in visually grounded tasks. As conceptually depicted in Figure 1 (bottom), both our methods steer the LVLM to balance its focus between the visual and textual modalities during training. Intuitively, as shown in the bottom panel of Figure 2, our LBR method enables the LVLM to sustain robust attention on the visual input, thereby mitigating language bias and hallucinations.

Extensive experiments conducted across multiple models and benchmarks provide strong evidence for the effectiveness of both LBR and LBP. Specifically, LBR yields consistent performance gains across more than ten general-purpose benchmarks, while LBP substantially improves model robustness and trustworthiness on multiple hallucination-focused evaluations.

Furthermore, our targeted human evaluation confirms that both LBR and LBP effectively mitigate language bias and the resulting hallucinations. Together, these results not only corroborate the efficacy of our proposed methods but also lend empirical support to our analysis regarding the role of language bias in LVLMs.

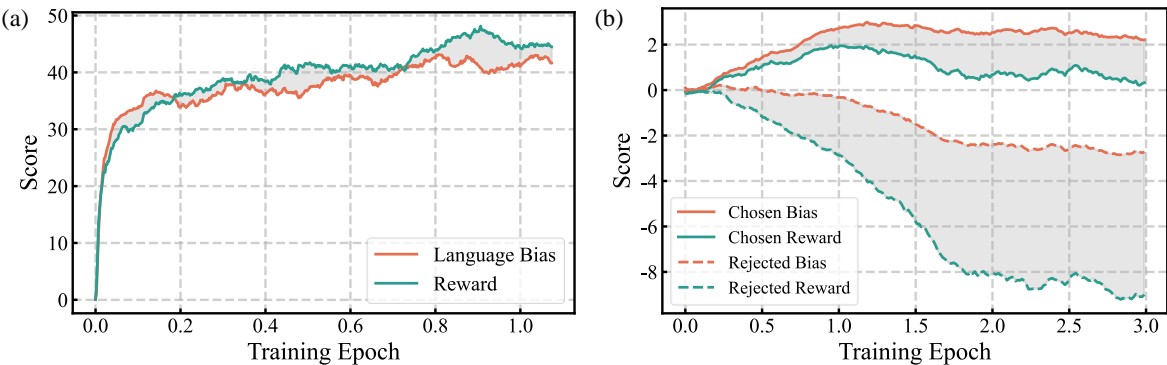

*Figure 3.* Evolution of language bias and reward (defined in Eq. 5) during the training processes of (a) Visual Instruction Tuning (VIT) and (b) Direct Preference Optimization (DPO).

In summary, our contributions are threefold:

- We present a novel perspective on modality misalignment in LVLMs. Through rigorous quantitative analysis, we uncover the phenomenon of *language bias*, where LVLMs neglect visual information and over-rely on internal language priors during multimodal training.
- We introduce **Language Bias Regularization (LBR)** and **Language Bias Penalty (LBP)**, two versatile and easily deployable methods that require neither additional models nor external data. Each method can be seamlessly integrated into its respective stage—LBR into visual instruction tuning and LBP into DPO training—to effectively mitigate language bias.
- We conduct extensive evaluations across models of varying scales and a wide range of datasets, demonstrating the effectiveness and strong generalization capability of LBR and LBP. The results confirm that our methods significantly improve vision-language alignment in LVLMs, while also providing empirical validation for our analysis of language bias.

## 2. Related Work

**Language Bias and Hallucination in LVLMs.** *Hallucination* (Sun et al., 2024; Zhou et al., 2024; Huang et al., 2024a; Bai et al., 2024), a phenomenon where the model generates linguistically fluent yet visually inconsistent descriptions of image content, stands as a critical challenge in modern LVLMs. This issue is widely attributed to *language bias* (Jiang et al., 2025): the tendency of models to prioritize linguistic patterns over visual faithfulness. While numerous mitigation strategies have been proposed, spanning instruction fine-tuning (Liu et al., 2024a; Jiang et al., 2024a), preference learning (Yu et al., 2024b;c; Yang et al., 2025), and improved decoding methods (Leng et al., 2024; Li et al., 2025), the underlying mechanisms of language bias remain poorly understood. Specifically, current understanding is predominantly qualitative; the field lacks a formal definition and the rigorous quantitative analysis required to

address the problem at its core.

## 3. In-Depth Analysis of Language Bias

### 3.1. Preliminaries

**Visual Instruction Tuning (VIT).** Modern LVLMs undergo a two-stage training process: an initial **Pre-Training (PT)** for coarse alignment on large-scale image-text pairs, followed by Visual Instruction Tuning (VIT) for fine-grained alignment on high-quality instruction data. Our analysis focuses on the VIT stage, as pre-trained models are typically limited to generating a series of short, descriptive phrases and exhibit minimal language bias (Figure 10).

The VIT objective is to fine-tune the model autoregressively using Maximum Likelihood Estimation (MLE). Given an image $v$, an instruction $x$, and a response $y$, the loss function is defined as:

$$\mathcal{L}_{\text{VIT}} = -\sum_{t=1}^{n} \log \pi_\theta(y_t \mid x, v, y_{<t}), \qquad (1)$$

where $\theta$ represents the model parameters, $y_t$ is the token at timestep $t$, $y_{<t}$ are the preceding tokens, and $n$ is the total length of the response $y$.

**Direct Preference Optimization (DPO).** DPO (Rafailov et al., 2024) aligns models with human preferences directly from preference data, offering a more streamlined alternative to Reinforcement Learning from Human Feedback (RLHF) (Stiennon et al., 2020; Ouyang et al., 2022), which often requires a separate and complex reward model. In the multimodal setting, DPO utilizes a preference dataset $\mathcal{D}$ of tuples $(x, v, y_w, y_l)$, containing a prompt $x$, an image $v$, a preferred response $(y_w)$, and a less preferred one $(y_l)$. The optimization objective is:

$$\mathcal{L}_{\text{DPO}} = -\log \sigma \left( \beta \log \frac{\pi_\theta(y_w \mid x, v)}{\pi_{\text{ref}}(y_w \mid x, v)} - \beta \log \frac{\pi_\theta(y_l \mid x, v)}{\pi_{\text{ref}}(y_l \mid x, v)} \right),$$
$$(2)$$

where $\pi_{\text{ref}}$ is the reference model and $\beta$ controls the policy

divergence. However, DPO can suffer from training instability due to reward hacking. To enhance stability, we follow the approach of (Wang et al., 2024; Jiang et al., 2024b) and incorporate a margin loss.

$$\mathcal{L}_{\text{Margin}} = -\log \sigma \left( \beta \log \frac{\pi_\theta(y_w \mid x, v)}{\pi_{\text{ref}}(y_w \mid x, v)} \right). \quad (3)$$

The final objective combines both losses, which we adopt as our baseline:

$$\mathcal{L}_{\text{DPO}_M} = \mathcal{L}_{\text{DPO}} + \mathcal{L}_{\text{Margin}}. \quad (4)$$

For simplicity, we refer to this combined objective $\mathcal{L}_{\text{DPO}_M}$ as **DPO** throughout the rest of the paper unless specified.

### 3.2. Decompose The Training Process

In this section, we decompose the training process to quantitatively analyze the emergence of language bias. Our analysis begins with the VIT stage. We hypothesize that standard conditional probability-based training may lead the model to neglect visual tokens due to the inherent modality gap. To test this, we track two key quantities during LLaVA v1.5 7B's instruction tuning:

$$\mathcal{R}_{\text{VIT}} = \log \frac{\pi_\theta(y \mid x, v)}{\pi_{\text{ref}}(y \mid x, v)}, \quad \mathcal{B}_{\text{VIT}} = \log \frac{\pi_\theta(y \mid x)}{\pi_{\text{ref}}(y \mid x)}, \quad (5)$$

where $\pi_{\text{ref}}$ is the pre-VIT reference model. $\mathcal{R}_{\text{VIT}}$ (reward) measures the gain on the full multimodal input, while $\mathcal{B}_{\text{VIT}}$ (bias) measures the gain from text-only conditioning. As shown in Figure 3(a), their nearly identical trajectories are strong quantitative evidence that the model's improvement is text-driven, substantiating the presence of language bias.

Similarly, we extend this analysis to DPO. We track the corresponding multimodal gain ($\mathcal{R}$) and text-only gain ($\mathcal{B}$) for both the preferred ($y_w$) and rejected ($y_l$) responses in each preference pair. Our experiments on the RLHF-V dataset reveal a consistent trend (Figure 3(b)). Notably, the text-only gain for preferred responses ($\mathcal{B}_{\text{DPO}_w}$) even outpaces the multimodal gain ($\mathcal{R}_{\text{DPO}_w}$), reinforcing that preference learning can also exacerbate language bias at the expense of visual grounding. Additional analysis and visualizations of language bias dynamics during training are provided in Appendix D.1.

Motivated by these findings, we formally define *language bias* as:

$$\mathcal{B} = \log \frac{\pi_\theta(y \mid x)}{\pi_{\text{ref}}(y \mid x)}. \quad (6)$$

Intuitively, $\mathcal{B}$ quantifies the model's performance gain from text-only conditioning relative to a reference model. A high $\mathcal{B}$ value indicates a strong reliance on its internal language priors, diminishing the contribution of the visual modality. This bias is a common artifact of multimodal training paradigms that rely on conditional probabilities, such as VIT and DPO.

## 4. Simple and Effective Mitigation of Language Bias

Building on our quantitative formulation of language bias from Section 3, we introduce two simple yet highly effective mitigation strategies.

### 4.1. Language Bias Regularization for Visual Instruction Tuning

As established in our analysis, language bias is minimal after the pre-training stage. Therefore, the primary goal during Visual Instruction Tuning (VIT) is not to introduce complex new constraints, but simply to **mitigate the excessive growth of language bias**. This prevents the model from over-relying on the linguistic modality at the expense of visual grounding.

To this end, we propose **Language Bias Regularization (LBR)**, which directly penalizes the magnitude of the language bias term $\mathcal{B}$:

$$\mathcal{L}_{\text{LBR}} = \left| \mathcal{B} \right| = \left| \log \frac{\pi_\theta(y \mid x)}{\pi_{\text{ref}}(y \mid x)} \right|. \quad (7)$$

The overall VIT training objective is then updated to include this regularization term:

$$\mathcal{L}'_{\text{VIT}} = \mathcal{L}_{\text{VIT}} + \alpha \cdot \mathcal{L}_{\text{LBR}}, \quad (8)$$

where $\alpha$ is a hyperparameter controlling the regularization strength. By minimizing $\mathcal{L}_{\text{LBR}}$, we constrain the model's text-only output distribution ($\pi_\theta(y \mid x)$) to remain close to that of the reference model. This simple mechanism effectively suppresses linguistic drift during training and encourages the model to better utilize visual information.

### 4.2. Language Bias Penalty for Direct Preference Optimization

Unlike the VIT stage where language bias is nascent, DPO begins with a model that has already acquired language bias from prior instruction tuning. A mild regularizer like LBR is insufficient for this scenario; a more potent and targeted penalty is needed to actively suppress this existing bias.

Inspired by the DPO loss formulation, we propose the **Language Bias Penalty (LBP)**:

$$\mathcal{L}_{\text{LBP}} = -\log \sigma(\mathcal{B}) = -\log \sigma \left( \beta \cdot \log \frac{\pi_{\text{ref}}(y \mid x)}{\pi_\theta(y \mid x)} \right), \quad (9)$$

where $y$ can be either the chosen ($y_w$) or rejected ($y_l$) response. The updated DPO objective is:

$$\mathcal{L}'_{\text{DPO}} = \mathcal{L}_{\text{DPO}_M} + \gamma \cdot \mathcal{L}_{\text{LBP}}, \quad (10)$$

where $\gamma$ controls the penalty strength. Minimizing $\mathcal{L}_{\text{LBP}}$ actively pushes the language bias $\mathcal{B}$ towards negative values.

*Table 1.* Evaluation of our LBR on text-intensive and visual QA benchmarks.

| Model | Text-intensive Tasks | | | | Visual QA Tasks | | | |
|---|---|---|---|---|---|---|---|---|
| | VQA$^{Chart}$ | VQA$^{Text}$ | VQA$^{Info}$ | OCRBench | GQA | SQA$^I$ | VisWiz | RWQA |
| LLaVA-1.5-7B | 17.1 | 45.8 | 21.5 | 31.6 | 62.0 | 66.8 | 50.1 | **55.4** |
| **LBR (ours)** | **17.3** | **46.0** | **21.7** | **32.0** | **62.7** | **69.4** | **54.0** | 54.9 |
| LLaVA-1.5-13B | 17.2 | 48.0 | 23.7 | 34.0 | 63.4 | 71.5 | **55.6** | 54.4 |
| **LBR (ours)** | **17.4** | **48.1** | **23.9** | **34.4** | **63.6** | **72.1** | 55.1 | **55.2** |
| LLaVA-NEXT-3B | 21.1 | 56.1 | 25.0 | 36.8 | 61.9 | 71.1 | 50.5 | 55.0 |
| **LBR (ours)** | **21.4** | **57.9** | **25.9** | **37.2** | **62.4** | **71.5** | **54.5** | **55.5** |

*Table 2.* Evaluation of our LBR on general LVLM capabilities and image caption benchmarks.

| Model | General LVLM Benchmarks | | | | | | Image Caption Tasks | |
|---|---|---|---|---|---|---|---|---|
| | MME | MMB$_{en}$ | Seed$_i$ | MMMU | MMT | MMStar | CocoCap | TextCap |
| LLaVA-1.5-7B | 1490 | 64.9 | **66.2** | 35.7 | 47.4 | 33.6 | 110.6 | 98.4 |
| **LBR (ours)** | **1525** | **65.3** | 65.9 | **37.2** | **47.5** | **33.9** | **112.1** | **99.1** |
| LLaVA-1.5-13B | 1525 | 67.1 | 67.5 | 36.7 | 48.8 | 34.2 | 112.1 | 103.9 |
| **LBR (ours)** | **1527** | **67.3** | **68.0** | **37.9** | **49.5** | **35.6** | **112.3** | **105.2** |
| LLaVA-NEXT-3B | 1420 | 69.2 | 71.4 | **39.6** | 51.7 | 42.7 | 109.4 | 104.0 |
| **LBR (ours)** | **1424** | **69.4** | **71.5** | 38.8 | 51.7 | **44.7** | **111.5** | **105.8** |

This encourages the model to "unlearn" the bias acquired during VIT and strengthen its reliance on visual information. Crucially, the properties of the sigmoid function $\sigma(\cdot)$ prevent this penalty from becoming excessively large, thus maintaining training stability.

## 5. Experiments

### 5.1. Experimental Setup

**Models and Datasets.** Our experiments utilize the LLaVA-v1.5 (Liu et al., 2024b) (7B, 13B) and LLaVA-NEXT (3B) models. For **LBR** (VIT), we train all models on the official LLaVA-v1.5 instruction tuning dataset, a widely-recognized, **open-source** dataset for visual instruction tuning. For **LBP** (DPO), we use the LLaVA-v1.5 models and train on RLHF-V (Yu et al., 2024a) (5.7K pairs), supplemented by 1K and 10K pairs from VLFeedback (Li et al., 2024) for scalability analysis.

**Evaluation Benchmarks.** We evaluate **LBR** on a comprehensive suite of benchmarks spanning four key categories: General LVLM Benchmarks, Text-intensive Tasks, Visual QA Tasks, and Image Caption Tasks. For **LBP**, our evaluation focuses on three hallucination-centric benchmarks: MMHalBench, AMBER (which comprises both Generative and Discriminative tasks), and Object HalBench. Further benchmark details are in Appendix B.1.

**Baselines.** For LBR, we compare against the vanilla VIT baseline. For LBP, We first compare LBP with DPO variants such as V-DPO (Xie et al., 2024) and MFPO (Jiang et al., 2024b), as they use the same model and data, allowing for **direct comparison**. In additional experiments, we compare with vanilla DPO (with Margin Loss), which shares the same training process, data, and hyper-parameters as LBR,

but with a different learning objective. In addition, we also report results from several representative models and methods for reference. Further details are in Appendix B.2.

**Implementation.** For VIT training, we follow the official training configurations for all LLaVA models. Key hyperparameters for our methods are set consistently across experiments: the LBR strength $\alpha$ is $1 \times 10^{-5}$. For DPO training, we set $\beta = 0.1$ and the LBP strength $\gamma$ is 1. The 7B model is fully fine-tuned for 3 epochs, while the 13B model is trained for 4 epochs using LoRA (Hu et al., 2021). Additional implementation details can be found in Section B.3.

### 5.2. Main Results

**LBR Enhances General LVLM Capabilities.** As detailed in Tables 1 and 2, our LBR method shows consistent improvements across a wide range of tasks. Across the four major categories of benchmarks—general understanding, text-intensive VQA, visual reasoning, and image captioning—LBR surpasses the vanilla VIT baseline on the vast majority of metrics. This broad outperformance validates LBR's ability to mitigate language bias and foster superior multimodal alignment.

**LBP Improves LVLM Trustworthiness.** As shown in Table 3, LBP achieves SOTA performance on key hallucination benchmarks, including MMHalBench, AMBER, and Object HalBench. LBP consistently outperforms all baselines across both 7B and 13B models, with particularly strong gains on benchmarks requiring long-form generation. Notably, with just 5.7K preference pairs, our LBP-aligned LLaVA-v1.5-7B model matches or exceeds the performance of GPT-4V on two sub-tasks of AMBER and Object HalBench. This advantage is pronounced on the challenging MMHalBench, where LBP cuts the hallucination rate of

*Table 3.* Main results of benchmarks measuring **trustworthiness**, where our LBP is trained on RLHF-V. Note that only methods using the same base model and training data are directly comparable, with the best result in each group highlighted in bold.

| Model | MMHalBench | | Generative Task | | | | Discriminative Task | | Object HalBench | |
|---|---|---|---|---|---|---|---|---|---|---|
| | Score ↑ | HalRate ↓ | CHAIR$_s$ ↓ | Cover. ↑ | HalRate ↓ | Cog. ↓ | Acc. ↑ | F1 ↑ | CHAIR$_s$ ↓ | CHAIR$_i$ ↓ |
| *Referenced Results (Not Directly Comparable)* | | | | | | | | | | |
| –*Base Models* | | | | | | | | | | |
| LLaVA-v1.5-7B (Liu et al., 2024b) | 2.07 | 0.59 | 8.5 | 50.5 | 39.1 | 4.6 | 72.0 | 74.7 | 53.6 | 25.2 |
| LLaVA-v1.5-13B (Liu et al., 2024b) | 2.36 | 0.55 | 8.8 | 50.2 | 37.3 | 4.3 | 79.3 | 84.4 | 46.3 | 22.6 |
| GPT-4V (Achiam et al., 2023) | 3.49 | 0.28 | 4.6 | 67.1 | 30.7 | 2.6 | 83.4 | 87.4 | 13.6 | 7.3 |
| –*LLaVA-v1.5-7B-based Baselines* | | | | | | | | | | |
| CCA-LLaVA (Xing et al., 2024) | 1.92 | 0.62 | 8.1 | 45.9 | 32.1 | 4.1 | 77.7 | 81.9 | 46.7 | 23.8 |
| mDPO (Wang et al., 2024) | 2.39 | 0.54 | 4.4 | 52.4 | 24.5 | 2.4 | – | – | 35.7 | 9.8 |
| RLAIF-V (Yu et al., 2024c) | 2.95 | 0.32 | 3.0 | 50.3 | 16.1 | 1.0 | 76.8 | 84.5 | 10.5 | 5.2 |
| OPA-DPO (Yang et al., 2025) | 2.83 | 0.45 | 2.2 | 47.9 | 11.6 | 0.9 | – | – | 13.0 | 4.3 |
| –*LLaVA-v1.5-13B-based Baselines* | | | | | | | | | | |
| RLHF-V (Yu et al., 2024b) | 2.45 | 0.51 | 6.3 | 46.1 | 25.1 | 2.1 | 72.6 | 75.0 | 12.2 | 7.5 |
| HSA-DPO (Xiao et al., 2024) | 2.61 | 0.48 | 2.1 | 47.3 | 13.4 | 1.2 | 80.8 | 86.1 | 5.3 | 3.2 |
| HALVA (Sarkar et al., 2025) | 2.84 | 0.42 | 6.4 | 52.6 | 30.4 | 3.2 | – | 86.5 | – | – |
| AMP-MEG (Zhang et al., 2024) | 3.08 | 0.37 | 11.0 | 53.8 | 45.8 | 5.6 | 79.5 | 84.6 | 31.7 | 20.6 |
| *Directly Comparable Results* | | | | | | | | | | |
| –*LLaVA-v1.5-7B-based Baselines* | | | | | | | | | | |
| V-DPO (Xie et al., 2024) | 2.16 | 0.56 | 5.6 | 49.7 | 27.3 | 2.7 | – | 81.6 | – | – |
| MFPO (Jiang et al., 2024b) | 2.69 | 0.49 | 4.1 | **55.7** | 22.5 | 1.9 | – | – | 13.4 | 6.6 |
| **LBP (ours)** | **2.91** | **0.43** | **3.5** | 53.2 | **18.5** | **1.6** | 78.6 | **86.1** | 12.3 | **6.3** |
| –*LLaVA-v1.5-13B-based Baselines* | | | | | | | | | | |
| MFPO (Jiang et al., 2024b) | 2.94 | 0.42 | 3.4 | **56.1** | 19.4 | 1.4 | – | – | 11.4 | 4.6 |
| **LBP (ours)** | **3.01** | 0.42 | **3.3** | 51.5 | **16.6** | **1.3** | 77.0 | 85.4 | **10.7** | **4.2** |

*Table 4.* Additional experimental results for our LBP method on the LLaVA-v1.5-7B model, trained on the VLFeedback dataset.

| Model | MMHalBench | | Generative Task | |
|---|---|---|---|---|
| | Score ↑ | HalRate ↓ | CHAIR$_s$ ↓ | HalRate ↓ |
| VLFeedback 1K | | | | |
| DPO | 2.25 | 0.59 | 8.7 | 41.5 |
| DPO$_M$ | 2.39 | 0.56 | 8.9 | 41.1 |
| LBP | **2.42** | **0.54** | **8.0** | **37.9** |
| VLFeedback 10K | | | | |
| DPO | 2.68 | 0.55 | 6.3 | 35.6 |
| DPO$_M$ | 2.77 | 0.47 | 6.2 | 31.2 |
| LBP | **2.82** | **0.46** | **6.1** | **30.5** |

*Table 5.* Ablation study of our LBP method on the LLaVA-v1.5-7B and 13B models, trained on the RLHF-V dataset.

| Model | MMHalBench | | Generative Task | |
|---|---|---|---|---|
| | Score ↑ | HalRate ↓ | CHAIR$_s$ ↓ | HalRate ↓ |
| LLaVA-v1.5-7B | | | | |
| DPO | 2.11 | 0.66 | **3.0** | 21.6 |
| DPO$_M$ | 2.42 | 0.54 | 4.5 | 25.9 |
| LBP | **2.91** | **0.43** | 3.5 | **18.5** |
| LLaVA-v1.5-13B | | | | |
| DPO | 2.61 | 0.53 | **2.6** | 18.2 |
| DPO$_M$ | 2.83 | 0.46 | 3.8 | 19.7 |
| LBP | **3.01** | **0.42** | 3.3 | **16.6** |

*Table 6.* Ablation study on different regularization methods for LBR, conducted on LLaVA-v1.5 7B.

| Method | MME | SQA | MMStar | OCRBench | VQA$^{Text}$ | CocoCap |
|---|---|---|---|---|---|---|
| **L1 (LBR)** | **1525** | 69.4 | **33.9** | **32.0** | **46.0** | 112.1 |
| L1 mean | 1493 | 69.2 | 33.8 | **32.0** | 45.9 | **112.5** |
| KL | 1469 | 69.8 | 33.3 | 31.2 | 45.5 | 110.7 |
| Contrastive | 1501.2 | **69.9** | 33.8 | 31.7 | 45.6 | 111.8 |

LLaVA-v1.5-7B by 27%, a significant margin over competing methods. While LBP was primarily designed for long-form generation, since language bias is pronounced in longer responses, it remains competitive on short-response discriminative tasks.

**LBR and LBP Demonstrate Strong Generalization.** Both LBR and LBP show excellent generalization across different settings. For LBR, we validated its performance across three model sizes and two distinct architectures. For LBP, as shown in Table 4, we confirmed its robustness by testing on varying scales of preference data (1K and 10K samples from VLFeedback), with full results presented in Table 11. Crucially, both methods achieved these strong results without any changes to their respective hyperparameters ($\alpha$ and $\gamma$), highlighting the robustness of our proposed techniques.

### 5.3. Ablation Studies

**Ablation Study of LBP.** We conduct an ablation study to isolate the performance gains of LBP, comparing it directly against vanilla DPO and DPO$_M$ in Table 5 (detailed results

are in Table 12). The results show that LBP consistently outperforms both baselines across nearly all metrics. The exception is the CHAIR score on the AMBER Generative Task and Object HalBench. We attribute this anomaly to a flaw in the CHAIR metric, which scores outputs by matching generated object words to ground-truth objects. This scoring mechanism is susceptible to reward hacking; models can artificially inflate their CHAIR score by **repeatedly describing a few salient objects** in the image. A more thorough discussion of the CHAIR metric's limitations is provided in Section D.2.

**Alternative Regularization Methods for LBR.** To validate our choice of regularization for LBR, we investigated several

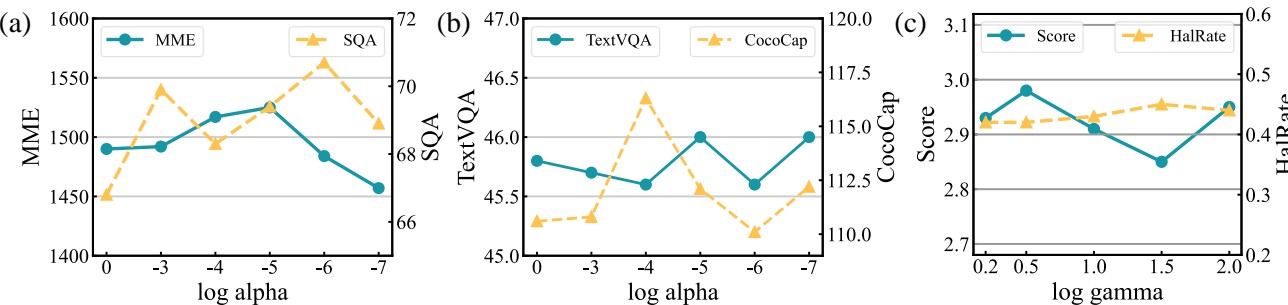

*Figure 4.* Ablation study on the hyperparameters for LBR and LBP. **(a, b)** Performance of our LBR method with varying values of hyperparameter $\alpha$. An x-axis value of 0 corresponds to the baseline where $\alpha = 0$. **(c)** Performance of our LBP method across different values of hyperparameter $\gamma$.

*Table 7.* Ablation of LBP on several general benchmarks.

| Model | GQA | MMBench$_{en}$ | MME | MMStar | MMT | SQA | VQA$^{Text}$ | VizWiz |
|-------|-----|-------------|------|--------|------|------|-----------|--------|
| LBP | 57.3 | 65.2 | 1117 | **34.7** | **48.6** | 68.2 | **44.6** | **53.3** |
| DPO$_M$ | **57.6** | **65.3** | **1126** | 34.5 | 48.4 | 68.0 | 44.5 | 53.1 |
| DPO | 55.5 | 64.7 | 1088 | 34.4 | 48.1 | 66.4 | 42.2 | 45.0 |

alternative strategies on the LLaVA-v1.5 7B model, beyond the proposed L1 penalty on sequence-level language bias. The alternatives included: (i) an L1 penalty on the *token-averaged* language bias (L1-Mean), (ii) a KL divergence constraint on the text-only output distribution (KL), and (iii) a DPO-style contrastive objective (Contrastive). Detailed implementation for each method is provided in Section B.4. As shown in Table 6, the proposed sequence-level L1 regularization yields the most stable and effective performance, confirming its selection as our final approach.

**Hyperparameter Sensitivity.** Our ablation studies in Figure 4 reveal that LBP is largely insensitive to its hyperparameter $\gamma$, while LBR exhibits greater sensitivity to its hyperparameter $\alpha$. We illustrate this by tracking the language bias dynamics during training under different hyperparameter settings, as visualized in Figure 7 and Section 6.4. For LBR, the plots show that the magnitude of $\alpha$ directly controls the regularization strength and onset; a larger $\alpha$ applies the constraint more forcefully, whereas a value as low as $1 \times 10^{-7}$ provides a negligible effect. In contrast, the penalty from LBP remains highly consistent across a range of $\gamma$ values, an observation that aligns with its stable performance on the MMHalBench benchmark.

# 6. Further Analysis

## 6.1. Impact of LBP on General Capabilities

A well-known challenge in preference learning is the trade-off between improving model trustworthiness and maintaining general capabilities; enhancing alignment often degrades performance on standard benchmarks. Our LBP method, however, successfully circumvents this issue. As shown in Table 7, applying LBP not only preserves the model's

overall performance but, in several cases, even improves it. This finding indicates that LBP's penalty mechanism is precisely targeted, allowing it to suppress *language bias* without causing collateral damage to the model's foundational abilities.

## 6.2. Human Evaluation of Language Bias

While our previous sections demonstrate broad performance gains, this section focuses on directly assessing the mitigation of *language bias* itself. We find that existing automated hallucination benchmarks are insufficient for this nuanced task, necessitating a targeted human study.

**Limitations of Existing Automated Hallucinatio Benchmarks.** A key finding of our study is the discrepancy between LBR's intended purpose—to mitigate language bias—and its measured performance on standard hallucination benchmarks. While designed to improve visual faithfulness, LBR shows only modest gains on benchmarks such as Object HalBench and MMHalBench (Section C.2 and Table 13). This discrepancy led us to hypothesize that existing benchmarks are ill-suited to capture the nuances of language bias. Specifically, a robust evaluation requires assessing whether a model can maintain its grounding in **fine-grained visual details** throughout the course of generating **long-form text**. Current benchmarks struggle to adequately test this sustained visual faithfulness, a limitation we discuss further in Section D.2.

**Human Evaluation Protocol.** To overcome the limitations of automated metrics, we conducted a human evaluation. We prompted the baseline, LBR, and LBP versions of **LLaVA-v1.5-7B** to generate detailed descriptions for 100 images randomly sampled from the COCO validation

**Prompt:** Please help me describe the image in detail.

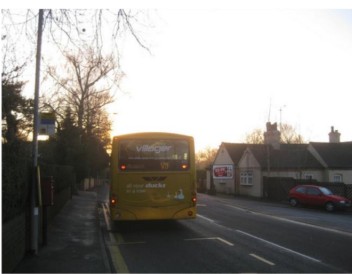

**Baseline**

The scene depicts a yellow bus driving down a street, with the sun setting in the background. The bus is positioned in the middle of the image, and it appears to be a public transit bus. There are no passengers visible on the bus. On the right side of the image, there are two cars parked on the side of the road. The first car is red and is located towards the right edge of the image, while the second car is parked further back and is also red. Additionally, there is a fire hydrant on the left side of the image, located near the bus.

**LBP**

The scene depicts a yellow bus driving down a street, with a red car parked on the right side of the street. The bus is positioned in the middle of the scene, and the car is located towards the right side of the image. There are some buildings visible in the background, with one of them having a sign on it. The sky can be seen in the background, and the sun is setting, creating a beautiful atmosphere.

*Figure 5.* A qualitative case study comparing our LBP-aligned model with the DPO baseline.

*Table 8.* Ablation on LBR regularization scheduling ($\alpha$) for LLaVA-1.5-7B, comparing Fixed vs. Cosine Annealing strategies across text-intensive and visual QA benchmarks.

| Method | Text-intensive Tasks | | | | Visual QA Tasks | | | |
|---|---|---|---|---|---|---|---|---|
| | VQA$^{\text{Chart}}$ | VQA$^{\text{Text}}$ | VQA$^{\text{Info}}$ | OCRBench | GQA | SQA$^{\text{I}}$ | VisWiz | RWQA |
| VIT (Baseline) | 17.1 | 45.8 | 21.5 | 31.6 | 62.0 | 66.8 | 50.1 | 55.4 |
| LBR (Fixed) | 17.3 | 46.0 | 21.7 | **32.0** | **62.7** | **69.4** | **54.0** | 54.9 |
| LBR (Cosine) | **17.4** | **46.1** | **21.8** | **32.0** | 62.4 | 69.3 | **54.0** | **55.6** |

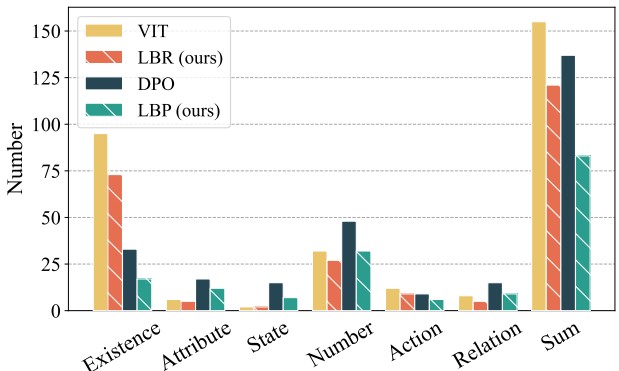

*Figure 6.* Human evaluation results evaluating fine-grained visual grounding. LBR and LBP significantly reduce hallucination rates across multiple categories compared to baselines, demonstrating robust mitigation of language bias.

dataset (Lin et al., 2014). Following the AMBER framework, human evaluators then assessed these descriptions for hallucinations across six dimensions (*Existence*, *Attribute*, *State*, *Number*, *Action*, and *Relation*) and recorded the number of errors. The full evaluation protocol is detailed in Section B.5.

**Results and Analysis.** The results of our human evaluation, presented in Figure 6, provide clear evidence of our methods' success. Both LBR and LBP are shown to effectively mitigate multiple types of hallucinations compared to the baseline. This outcome validates our core motivation: that explicitly targeting and reducing *language bias* is a potent strategy for enhancing the trustworthiness of LVLMs. Furthermore, we observe an interesting trade-off with the vanilla DPO model (relative to the VIT baseline). While it

significantly reduces *Existence*-type hallucinations, it concurrently increases hallucinations across nearly all other categories. This phenomenon serves as further evidence of language bias acquired during DPO training. It suggests the model is learning to mimic linguistic patterns in the preference data, rather than achieving a deeper, visually grounded understanding of the content.

### 6.3. Case Study

We present a representative case study in Figure 5 to qualitatively illustrate the benefits of LBP. The baseline model produces a description with several factual inaccuracies. It erroneously claims there are **two** cars, describes the position of a non-existent vehicle, and hallucinates a **fire hydrant** on the left. According to our evaluation protocol, these errors constitute one *Number* and one *Existence* hallucination. In contrast, the description generated by our LBP-aligned model is factually accurate and entirely free of such errors. This example vividly demonstrates LBP's effectiveness in suppressing object hallucinations and producing more trustworthy, visually grounded responses. Additional case studies of LBR and LBP are provided in Section E.

### 6.4. Training Dynamics and Dynamic Scheduling

To understand how our proposed methods mitigate language bias, we visualize the training dynamics by tracking the value of the language bias term, $\mathcal{B}$, throughout the training process. The experiments were conducted using the **LLaVA-v1.5-7B** model. As shown in Figure 7, we tracked the training dynamics of LBR (Figure 7 (a)) across several values for the hyperparameter $\alpha$, comparing them against

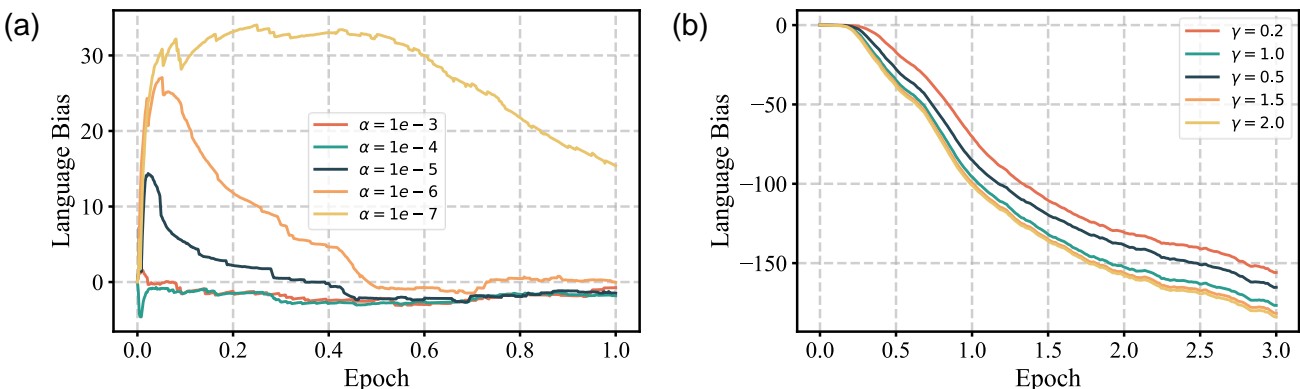

*Figure 7.* Language bias dynamics during training under different hyperparameter settings for LBR and LBP. **(a)** For LBR, the regularization effect is sensitive to the hyperparameter $\alpha$. **(b)** In contrast, the penalty from LBP is robust and remains consistent across a range of $\gamma$ values.

*Table 9.* Ablation study on regularization scheduling strategies ($\alpha$) for LLaVA-1.5-7B, on General LVLM and Image Captioning benchmarks.

| Method | General LVLM Benchmarks | | | | | | Image Caption Tasks | |
|---|---|---|---|---|---|---|---|---|
| | MME | MMB$_{en}$ | Seed$_i$ | MMMU | MMT | MM-Star | CocoCap | TextCap |
| VIT (Baseline) | 1490 | 64.9 | 66.2 | 35.7 | 47.4 | 33.6 | 110.6 | 98.4 |
| LBR (Fixed) | 1525 | **65.3** | 65.9 | **37.2** | 47.5 | **33.9** | 112.1 | 99.1 |
| LBR (Cosine) | **1526** | 64.9 | **66.4** | 36.9 | **47.8** | 33.6 | **113.3** | **100.2** |

the baseline where $\alpha = 0$. Similarly, we tracked the dynamics of LBP (Figure 7 (b)) across a range of values for the hyperparameter $\gamma$.

Insights from these training dynamics reveal that while a larger $\alpha$ in LBR accelerates early bias suppression, maintaining such a high penalty may lead to over-constraint as the model converges. To balance this, we further investigate a dynamic scheduling strategy for the regularization weight $\alpha$. Specifically, we employ a **Cosine Annealing** schedule, initializing $\alpha$ at $1 \times 10^{-4}$ and decaying it to $1 \times 10^{-6}$.

As shown in Tables 8 and 9, this dynamic approach (**LBR (Cosine)**) achieves superior performance across most benchmarks compared to the fixed-weight baseline. These results underscore the strong potential of our method and highlight adaptive regularization as a promising direction for future research. However, to maintain the narrative flow, we do not elaborate further on other dynamic scheduling algorithms in this work.

Furthermore, we provide additional detailed analysis and extended experiments in Appendix D.

## 7. Conclusion

In this work, we systematically analyze *language bias* in LVLMs, tracing the model's over-reliance on its language modality to a core misalignment in training dynamics, where processes like VIT and DPO often prioritize textual improvements over visual alignment. Based on this finding, we propose two targeted interventions: **Language Bias Regularization (LBR)** for VIT and **Language Bias Penalty (LBP)** for DPO. Experiments demonstrate that these simple training modifications consistently improve general capabilities (LBR) and significantly reduce hallucinations (LBP) without introducing any additional data or auxiliary models. Ultimately, this work offers both a deeper understanding of language bias and a practical path toward more reliable and aligned LVLMs.

## Impact Statement

This paper presents work whose goal is to advance the field of Machine Learning, specifically focusing on improving the reliability and visual faithfulness of Large Vision-Language Models (LVLMs). By mitigating language bias and reducing hallucinations, our proposed methods (LBR and LBP) contribute to the development of safer and more trustworthy AI systems. This is particularly critical for real-world applications where model factuality is paramount. We do not foresee any immediate negative societal consequences specific to this work, beyond the general risks associated with the advancement of generative AI models.

## Acknowledgements

This work was supported in part by National Natural Science Foundation of China (62476070), Shenzhen Science and Technology Program (JCYJ20241202123503005,

GXWD20231128103232001, ZDSYS20230626091203008, KQTD20240729102154066), Department of Science and Technology of Guangdong (2024A1515011540), National Key R&D Program of China (SQ2024YFE0200592) and Suzhou Science and Technology Program (SYG2025072).

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

## A. The Use of Large Language Models

Throughout the preparation of this manuscript, large language models were employed exclusively for light stylistic refinement, translation, and occasional grammatical adjustments. Every conceptual insight, analytical method, and interpretive conclusion originated solely from the authors; no algorithmic assistance was used for the framing, design, or substance of the scientific work. Full responsibility for the content and its claims rests with the human authors alone.

## B. Additional Experimental Setups

### B.1. Details of Benchmarks

We evaluate **LBR** on a comprehensive suite of benchmarks spanning four key categories: General LVLM Benchmarks, Text-intensive Tasks, Visual QA Tasks, and Image Caption Tasks.

**Text-intensive Tasks.** In Table 1, we use abbreviations for the following benchmarks, which test a model's ability to understand and reason about dense text within images:

- **TextVQA (VQA$^{\text{Text}}$)** (Singh et al., 2019): A benchmark that requires models to read and comprehend text in images to answer questions. We report results on the *validation* split.

- **ChartQA (VQA$^{\text{Chart}}$)** (Masry et al., 2022): A visual question answering dataset focused on understanding and reasoning about chart images. Evaluation is performed on the human-generated subset of the *test* split.

- **InfographicVQA (VQA$^{\text{Info}}$)** (Mathew et al., 2022): A dataset for VQA on infographics, which contain complex layouts, diverse text, and rich visual elements. We evaluate using the official *val* split.

- **OCRBench** (Liu et al., 2023b): A comprehensive benchmark designed to evaluate a model's optical character recognition (OCR) and text-centric visual understanding capabilities across a wide variety of scenarios. We report scores on its official *test* set.

**Visual QA Tasks.** In Table 1, we use abbreviations for the following benchmarks:

- **GQA** (Hudson & Manning, 2019): A visual reasoning and compositional question answering benchmark built on real-world images and their associated scene graphs. We report scores on the *test-dev-balanced* split.

- **VizWiz** (Gurari et al., 2018): A visual question answering dataset sourced from questions posed by blind and visually impaired individuals about their surroundings. We evaluate on the *validation* split.

- **ScienceQA (SQA$^{\text{I}}$)** (Lu et al., 2022): A large-scale multimodal benchmark featuring multiple-choice science questions derived from elementary to high school curricula. We evaluate on the subset of questions that include image context (**SQA$^{\text{I}}$**) using the *test* split.

- **RealWorldQA (RWQA)** (xAI, 2024): RealWorldQA is a benchmark designed for real-world understanding. The dataset consists of anonymized images taken from vehicles, in addition to other real-world images. We report results on the official *test* split.

**General LVLM Benchmarks.** In Table 2, we use abbreviations for the following benchmarks, which are designed to provide a comprehensive evaluation of a model's core multimodal capabilities:

- **MME** (Fu et al., 2023): A comprehensive benchmark designed to evaluate both the perception and cognition abilities of LVLMs across 14 different sub-tasks. We report the sum of perception scores.

- **MMBench (MMB$_{en}$)** (Liu et al., 2024d): A multi-dimensional benchmark that evaluates core multimodal capabilities such as perception, reasoning, and attribute recognition using a circular evaluation strategy. We report results on the English version using the *dev* split.

- **SEED-Bench (Seed$_i$)** (Li et al., 2023): A benchmark designed to assess fine-grained multimodal understanding across 12 evaluation dimensions, such as identifying attributes, scenes, and relationships. We evaluate on the image-based version (**Seed$_i$**).

*Table 10.* Hyper-parameter settings of LBP training.

| Hyper-parameters | LLaVA-v1.5-7B | LLaVA-v1.5-13B |
|---|---|---|
| Epoch | 3 | 4 |
| Learning rate | 5e-7 | 1e-6 |
| Batch size | 8 | 8 |
| Optimizer | AdamW | AdamW |
| Weight decay | 0.01 | 0.01 |
| Warmup ratio | 0.05 | 0.05 |
| $\beta$ | 0.1 | 0.1 |
| Bfloat16 | True | True |
| LoRA enable | False | True |
| LoRA $\alpha$ | – | 256 |
| LoRA rank | – | 128 |

- **MMMU** (Yue et al., 2024): A massive, multi-discipline benchmark that requires expert-level, college-exam-grade knowledge to answer questions spanning six core disciplines, from science and engineering to art and design. We report results on the *validation* set.

- **MMT-Bench (MMT)** (Ying et al., 2024): A benchmark specifically designed for evaluating multi-turn multimodal conversation and instruction-following capabilities. We report scores on its official *val* set.

- **MM-Star** (Chen et al., 2024a): A challenging benchmark featuring a wide array of advanced multimodal capabilities, including coarse- and fine-grained perception, logical reasoning, and resilience to difficult negative examples. We report the average score across all sub-tasks.

**Image Caption Tasks.** In Table 2, we use abbreviations for the following benchmarks, which evaluate the model's ability to generate descriptive text for images:

- **COCO Captions (CocoCap)** (Chen et al., 2015): The standard benchmark for image captioning on everyday scenes, based on the COCO (Common Objects in Context) dataset. We report the CIDEr score on the *val* split.

- **TextCap** (Sidorov et al., 2020): A challenging captioning benchmark where models must read and incorporate textual information present in the image to generate a coherent description. We report the CIDEr score on the *val* split.

For **LBP**, our evaluation focuses on the following three **hallucination benchmarks**:

- **MMHalBench** (Sun et al., 2024): Following the official protocol, we use GPT-4 ('gpt-4-0613') to assess the overall quality of generated responses on a scale from 0 to 6 and to calculate the final hallucination rate.

- **AMBER** (Wang et al., 2023): This benchmark consists of two parts. For the **Discriminative Task**, we report Accuracy and F1 scores. For the **Generative Task**, we use the official evaluation tool to report a CHAIR score variant, object coverage, rate of hallucinated responses, and hallucination rate overlapping with human cognition.

- **Object HalBench** (Rohrbach et al., 2018): Following prior work (Yu et al., 2024b; Wang et al., 2024; Fu et al., 2025), we report both the response-level ($CHAIR_s$) and mention-level ($CHAIR_i$) hallucination rates. Object extraction for this metric is performed using GPT-3.5-Turbo ('gpt-3.5-turbo-0125').

### B.2. Details of Baselines

For LBR, we also provide results from other MLLMs and methods for reference, which are not directly comparable due to differences in base models and preference data. These methods and models include GPT-4V (Achiam et al., 2023), RLAIF-V (Yu et al., 2024c), CCA-LLaVA (Xing et al., 2024), mDPO (Wang et al., 2024), RLHF-V (Yu et al., 2024b), HSA-DPO (Xiao et al., 2024), AMP-MEG (Zhang et al., 2024), HALVA (Sarkar et al., 2025), and OPA-DPO (Yang et al., 2025).

## B.3. Implementation Details

All experiments were conducted on a single server equipped with eight NVIDIA A800-SXM4-80GB GPUs. To ensure computational efficiency and minimize resource requirements, we pre-computed and cached the outputs of the reference model before starting our main training runs. This strategy allows both our LBR and LBP methods to train with VRAM usage nearly identical to that of the baseline, incurring negligible memory overhead and only a minor increase in training time.

**LBR Implementation.** For the Visual Instruction Tuning stage with LBR, we followed the official training procedure of LLaVA. Training the 7B model required 4 GPUs and took approximately 30 hours, while the 13B model utilized 8 GPUs and took about 36 hours. Given that our setup closely mirrors the standard LLaVA training, we omit a detailed reiteration of common hyperparameters.

**LBP Implementation.** For the Direct Preference Optimization stage with LBP, all experiments were conducted using 4 GPUs. On the RLHF-V dataset, fine-tuning the LLaVA-v1.5-7B model for 3 epochs took approximately 1.4 hours. Fine-tuning the LLaVA-v1.5-13B model for 4 epochs required about 2.6 hours. Training times on the larger VLFeedback dataset scale proportionally with the data size. A comprehensive list of all hyperparameters for LBP is provided in Table 10.

## B.4. Regularization Method Implementation

In our ablation study (Table 6), we compared our proposed sequence-level L1 regularization, $\mathcal{L}_{\text{LBR}} = |\mathcal{B}|$, against three alternative strategies. Below are the detailed formulations for these alternatives.

**1. Token-Averaged L1 Regularization (L1-Mean).** This approach normalizes the language bias by the length of the generated sequence before applying the L1 penalty. The intuition is to regularize the average language bias per token rather than the cumulative language bias of the entire sequence. The loss is defined as:

$$\mathcal{L}_{\text{LBR}_{\text{mean}}} = \left| \frac{1}{|y|} \sum_{t=1}^{|y|} \log \frac{\pi_\theta(y_t \mid x, y_{<t})}{\pi_{\text{ref}}(y_t \mid x, y_{<t})} \right| = \left| \frac{1}{|y|} \log \frac{\pi_\theta(y \mid x)}{\pi_{\text{ref}}(y \mid x)} \right|. \tag{11}$$

**2. KL Divergence Constraint (KL).** This method constrains the text-only output distribution of the current model, $\pi_\theta(y|x)$, to remain close to that of the reference model, $\pi_{\text{ref}}(y|x)$. Instead of the standard KL divergence, we use a penalty function derived from a Taylor approximation of the reverse KL divergence. This provides a stable and effective constraint. Let $d = \log \pi_{\text{ref}}(y \mid x) - \log \pi_\theta(y \mid x)$; the loss is then defined as:

$$\mathcal{L}_{\text{LBR}_{\text{KL}}} = \exp(d) - d - 1. \tag{12}$$

**3. DPO-Style Contrastive Objective (Contrastive).** Inspired by Direct Preference Optimization, this objective reframes the task as encouraging a positive margin between the full multimodal likelihood and the text-only likelihood. It aims to ensure that the gain from adding visual information is maximized. The loss function is defined as:

$$\mathcal{L}_{\text{LBR}_{\text{contrastive}}} = -\log \sigma \left( \log \frac{\pi_\theta(y \mid x, v)}{\pi_{\text{ref}}(y \mid x, v)} - \log \frac{\pi_\theta(y \mid x)}{\pi_{\text{ref}}(y \mid x)} \right), \tag{13}$$

where $\sigma(\cdot)$ is the sigmoid function.

**Hyperparameter Selection.** For each of the alternative methods above, we performed a series of validation experiments to select an appropriate weighting hyperparameter. Based on these experiments, the final weights used for the L1-Mean, KL, and Contrastive objectives in our ablation study (Table 6) were set to $1 \times 10^{-2}$, $1 \times 10^{-4}$, and $1 \times 10^{-1}$, respectively.

## B.5. Human Evaluation Setup

To provide a more nuanced assessment of language bias and its mitigation, we designed and executed a human evaluation study. The protocol was structured as follows:

**1. Stimuli and Task Definition.**  We randomly sampled 100 images from the COCO 2014 validation split. For each image, the models were given a single, open-ended instruction: `"Please help me describe the image in detail"`. This prompt was chosen to encourage the generation of long-form, descriptive text, which provides a rich context for identifying potential hallucinations.

**2. Models.**  We compared three versions of the **LLaVA-v1.5-7B** model:

- **Baseline:** The standard model after completing Visual Instruction Tuning (VIT) and Direct Preference Tuning (DPO).

- **LBR (ours):** The baseline model trained with our Language Bias Regularization.

- **LBP (ours):** The baseline model trained with our Language Bias Penalty.

**3. Hallucination Taxonomy and Annotation Rules.**  Our evaluation is based on a detailed hallucination taxonomy. Three trained human annotators were tasked with identifying and categorizing errors in the generated text based on the visual evidence. We categorize hallucinations into six types:

- **Existence:** The model describes objects that do not exist in the image.

- **Attribute:** Incorrect properties of an object (e.g., color or size) are hallucinated.

- **State:** The condition or status of an object is incorrectly described (e.g., "open" vs. "closed").

- **Number:** The count of objects is inaccurately stated.

- **Action:** Actions or activities that are not occurring are mistakenly attributed to objects.

- **Relation:** False spatial or semantic relationships between objects are generated.

To ensure consistency in annotation, we established the following rules:

1. For an initial **Existence** hallucination, any subsequent errors concerning the same non-existent object (e.g., its attributes, state, or relations) are not counted as additional hallucinations to avoid penalizing cascading errors.

2. Descriptions that explicitly convey uncertainty (e.g., "it seems like", "there might be") without introducing a concrete factual error are not considered hallucinations.

The final error count for each generated response was determined by a majority vote among the annotators.

## C. Detailed Experimental Results

### C.1. Detailed Experimental Results for LBP

#### C.1.1. DETAILED RESULTS OF ADDITIONAL EXPERIMENTS

Table 11 presents the comprehensive experimental results on the VLFeedback dataset, serving as the extended version of Table 4 from the main paper. While the main text demonstrated the efficacy of our method on 1k and 10k subsets, here we further verify its scalability by conducting an additional experiment using a larger **30k subset**.

As shown in the table, LBP consistently outperforms both the standard DPO and the Modified DPO (DPO$_M$) baselines on this larger scale. Specifically, LBP maintains a clear advantage on hallucination-centric benchmarks (MMHalBench, AMBER Generative Task, and Object HalBench) as the dataset size increases, while achieving comparable performance on the Discriminative Task. These results confirm that our method scales effectively with data size, robustly maintaining its advantage over competitive baselines.

*Table 11.* Detailed experimental results for our LBP method on the LLaVA-v1.5-7B model, trained on the VLFeedback dataset with varying data scales (1K, 10K, 30K). This table provides a complete version of the results in Table 4.

| Model | MMHalBench | | Generative Task | | | | Discriminative Task | | Object HalBench | |
|---|---|---|---|---|---|---|---|---|---|---|
| | Score ↑ | HalRate ↓ | CHAIR$_s$ ↓ | Cover. ↑ | HalRate ↓ | Cog. ↓ | Acc. ↑ | F1 ↑ | CHAIR$_s$ ↓ | CHAIR$_i$ ↓ |
| VLFeedback 1K | | | | | | | | | | |
| DPO | 2.25 | 0.59 | 8.7 | 51.4 | 41.5 | 5.0 | **78.2** | **83.9** | 56.0 | 28.5 |
| DPO$_M$ | 2.39 | 0.56 | 8.9 | **51.5** | 41.1 | 5.2 | **78.2** | **83.9** | 55.0 | 29.1 |
| LBP | **2.42** | **0.54** | **8.0** | 51.4 | **37.9** | **4.6** | 77.8 | 83.5 | **54.5** | **27.2** |
| VLFeedback 10K | | | | | | | | | | |
| DPO | 2.68 | 0.55 | 6.3 | **53.9** | 35.6 | 3.4 | 77.7 | 85.6 | 42.4 | 22.8 |
| DPO$_M$ | 2.77 | 0.47 | 6.2 | 53.0 | 31.2 | 3.6 | **80.1** | **86.3** | 38.3 | 21.2 |
| LBP | **2.82** | **0.46** | **6.1** | 53.0 | **30.5** | **3.0** | **80.1** | **86.3** | **37.1** | **20.9** |
| VLFeedback 30K | | | | | | | | | | |
| DPO | 2.80 | 0.54 | 5.8 | **53.5** | 31.9 | **2.7** | 79.5 | **85.9** | 33.5 | 18.5 |
| DPO$_M$ | 3.02 | 0.41 | 5.7 | 50.5 | 27.9 | 2.8 | 79.8 | 85.8 | 31.2 | 17.5 |
| LBP | **3.05** | **0.39** | **5.4** | 51.1 | **26.6** | **2.7** | **79.9** | **85.9** | **30.1** | **17.1** |

*Table 12.* Detailed ablation study of our LBP method on the LLaVA-v1.5-7B and 13B models, trained on the RLHF-V dataset. This table presents the complete results corresponding to Table 5.

| Model | MMHalBench | | Generative Task | | | | Discriminative Task | | Object HalBench | |
|---|---|---|---|---|---|---|---|---|---|---|
| | Score ↑ | HalRate ↓ | CHAIR$_s$ ↓ | Cover. ↑ | HalRate ↓ | Cog. ↓ | Acc. ↑ | F1 ↑ | CHAIR$_s$ ↓ | CHAIR$_i$ ↓ |
| LLaVA-v1.5-7B | | | | | | | | | | |
| DPO | 2.11 | 0.66 | **3.0** | 52.7 | 21.6 | **1.0** | **78.7** | 86.0 | **11.0** | **5.2** |
| DPO$_M$ | 2.42 | 0.54 | 4.5 | **53.6** | 25.9 | 1.7 | 78.3 | 86.0 | 15.8 | 8.0 |
| LBP | **2.91** | **0.43** | 3.5 | 53.2 | **18.5** | 1.6 | 78.6 | **86.1** | 12.3 | 6.3 |
| LLaVA-v1.5-13B | | | | | | | | | | |
| DPO | 2.61 | 0.53 | **2.6** | 51.0 | 18.2 | **0.7** | 76.2 | 84.9 | **9.0** | 4.5 |
| DPO$_M$ | 2.83 | 0.46 | 3.8 | **52.2** | 19.7 | 1.7 | 76.4 | 85.1 | 14.6 | 7.8 |
| LBP | **3.01** | **0.42** | 3.3 | 51.5 | **16.6** | 1.3 | **77.0** | **85.4** | 10.7 | **4.2** |

### C.1.2. DETAILED RESULTS OF ABLATION EXPERIMENTS

Table 12 shows the detailed ablation results using the RLHF-V dataset as the training data, which is the full version of Table 5. Consistent with our previous findings, LBP outperforms DPO most benchmarks, except on the the CHAIR-related benchmarks, which suffers from evaluation limitations. We provide a detailed discussion of the limitations of the CHAIR metric in Section D.2.

### C.2. LBR Evaluation on Automated Hallucination Benchmarks

The results of our evaluation on automated hallucination benchmarks are presented in Table 13. The table shows that our LBR method consistently outperforms the baseline on both Object HalBench and MMHalBench, although the numerical gains are modest.

## D. Extended Experiments and Supplementary Analysis

### D.1. Additional Analysis of Language Bias Dynamics

To demonstrate the pervasiveness of language bias across diverse datasets and model architectures, we provide additional training dynamics in Figure 8. Specifically, we plot the trajectories of Language Bias and Reward during DPO for (a) LLaVA-v1.5-7B trained on the VLFeedback dataset and (b) Qwen2.5-VL-3B trained on the RLHF-V dataset. The definitions for all metrics remain consistent with those in Figure 3 (b).

The results confirm that the emergence of language bias is a common phenomenon in DPO training. However, for the more advanced Qwen2.5-VL-3B model (Figure 8 (b)), we observe a more pronounced gap between the multimodal reward ($\mathcal{R}$) and the text-only bias ($\mathcal{B}$) for chosen responses compared to LLaVA-v1.5. This indicates that while language bias persists, the more advanced architecture exhibits a relatively lower degree of reliance on pure language priors.

*Table 13.* Object HalBench and MMHalBench evaluation for LBR.

| Method | Object HalBench | | MMHalBench | |
|---|---|---|---|---|
| | CHAIR$_s$ ↓ | CHAIR$_i$ ↓ | Score ↑ | HalRate ↓ |
| **LBR (ours)** | **52.2** | **26.7** | **2.10** | **0.57** |
| LLaVA-1.5-7B | 54.7 | 27.6 | 2.07 | 0.59 |

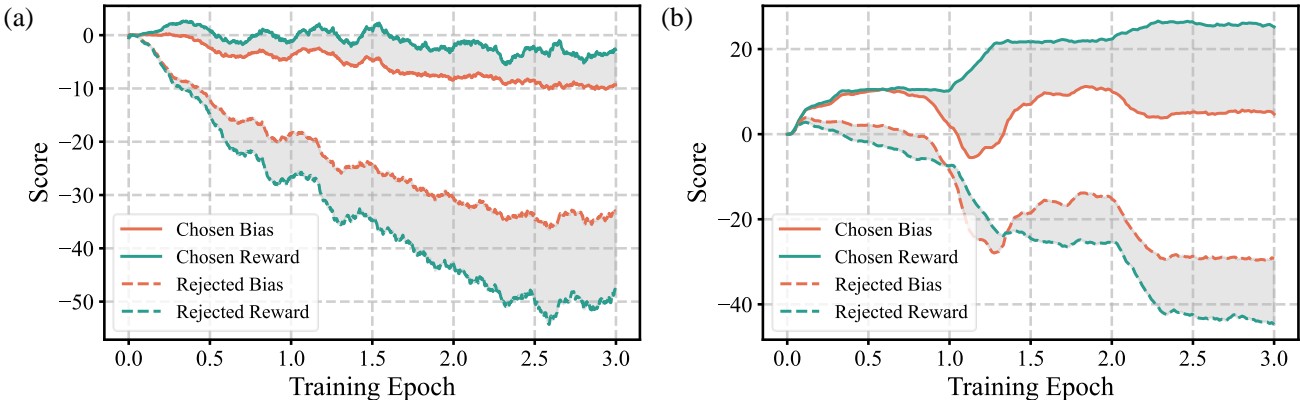

*Figure 8.* Evolution of Language Bias and reward during DPO training across different setups. **(a)** LLaVA-v1.5-7B trained on the VLFeedback dataset. **(b)** Qwen2.5-VL-3B trained on the RLHF-V dataset. The metrics follow the definitions provided in Section 3.

## D.2. Limitations of Automated Hallucination Benchmarks

Current automated benchmarks for hallucination primarily fall into two categories, both of which possess significant limitations: those based on object-matching and those using powerful Large Language Models (LLMs) as judges.

**1. Object-Matching Metrics (e.g., CHAIR).** Many benchmarks, including the generative tasks in AMBER (Wang et al., 2023) and Object HalBench (Rohrbach et al., 2018), rely on metrics like CHAIR. This approach operates by calculating the lexical overlap between object words in a generated caption and a pre-defined list of ground-truth objects. While straightforward, this method suffers from two fundamental flaws:

- **Incomplete Ground Truth:** Annotations are often incomplete, leading to false positives where correctly identified objects are penalized simply because they are missing from the ground-truth list. As illustrated in Figure 9, a model might accurately describe a "stove" and a "bottle", yet have them flagged as hallucinations because the ground truth only contains "orange" and "person".

- **Inability to Assess Relational Errors:** By focusing only on individual object words, these metrics cannot detect more complex errors in attributes, states, or the spatial and semantic relationships between objects.

**2. LLM-as-Judge Methods (e.g., MMHalBench).** More recent benchmarks like MMHalBench (Sun et al., 2024) leverage powerful LLMs (e.g., GPT-4) as judges to provide a more nuanced, semantic evaluation. While this approach can better assess overall coherence and relational reasoning compared to simple lexical matching, it is not without its own defects. The core issue is that the LLM judge does not perform a direct analysis of the image. Instead, it typically compares the generated text against ground-truth captions or annotations. This means the evaluating LLM lacks genuine visual grounding and can still fail to detect subtle visual inconsistencies or be misled by descriptions that are linguistically plausible but factually incorrect with respect to the image.

Given that both major types of automated metrics have inherent limitations, we concluded that a fine-grained human evaluation study (Section 6.2) was necessary to directly and reliably assess the impact of *language bias* on model-generated content.

**Prompt:** Please provide a detailed description of the image, including its visual elements, such as colors, shapes, textures, objects, and people.

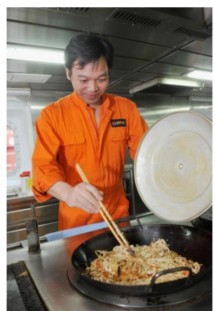

**Model output**

The image features a **man** wearing an **orange** shirt, standing in a kitchen and preparing food. He is using chopsticks to cook noodles in a wok, which is placed on a **stove**. The man appears to be focused on his task.

In addition to the man and the wok, there are two **bottles** visible in the scene. One bottle can be seen on the left side of the image, and the other is located on the left edge of the picture. The presence of these bottles suggests that the man might be using them for cooking purposes.

**Ground Truth Words**

"orange", "person"

**"Hallucinated" Words**

"stove", "bottle"

*Figure 9.* Illustration of the limitations of the CHAIR metric. The model provides a factually correct description, but objects like "stove" and "bottle" are penalized as hallucinations due to incomplete ground-truth object annotations.

*Table 14.* Experimental results generalizing LBP to the Qwen2.5-VL-3B model trained on the RLHF-V dataset. While the base model is already highly optimized, LBP consistently surpasses standard DPO under identical training conditions.

| Model | MMHalBench | | Generative Task | | | | Discriminative Task | | Object HalBench | |
|---|---|---|---|---|---|---|---|---|---|---|
| | Score ↑ | HalRate ↓ | CHAIR$_s$ ↓ | Cover. ↑ | HalRate ↓ | Cog. ↓ | Acc. ↑ | F1 ↑ | CHAIR$_s$ ↓ | CHAIR$_i$ ↓ |
| Qwen2.5-VL-3B | **3.28** | **0.44** | 8.5 | **69.5** | 52.8 | 6.0 | **81.5** | **86.5** | **13.4** | 7.7 |
| DPO | 2.98 | 0.49 | 5.7 | 58.2 | 30.3 | 2.2 | 81.0 | 86.4 | 17.3 | 8.7 |
| LBP (Ours) | 3.07 | 0.46 | **5.4** | 58.0 | **28.3** | **2.0** | 81.2 | 86.4 | 17.0 | **7.6** |

## D.3. Generalization to State-of-the-Art Architectures

To verify scalability, we applied LBP to **Qwen2.5-VL-3B** on the RLHF-V dataset, maintaining the same setup as our main experiments. It is important to note that we restricted this extension to LBP, as the implementation of LBR requires access to the data and model checkpoints from the visual instruction tuning stage, which are not publicly available for the Qwen series. As shown in Table 14, **LBP consistently outperforms standard DPO** across benchmarks, confirming effective generalization to the Qwen architecture. Note that additional training does not universally yield gains over the *Base* model, which is expected given its extensive prior alignment (including DPO). However, the critical takeaway is that **under identical training conditions, LBP is superior to DPO**. This robustly validates the effectiveness of our penalty term, even when applied to advanced, highly optimized architectures.

## D.4. Robustness of LBP in Long-form Generation

To rigorously quantify the impact of response length on hallucination rates, we extended the AMBER Generative Task by employing targeted prompts designed to induce long-form outputs, allowing for a stratified assessment across increasing token length buckets. As detailed in Table 15, while LBP and the DPO baseline exhibit comparable performance in short-context scenarios (< 64 tokens), a distinct divergence emerges as the response length increases; specifically, in long-context scenarios (128+ tokens), the hallucination rate for DPO escalates sharply (e.g., reaching 24.6 at length 128), whereas LBP significantly suppresses this upward trend (maintaining the rate at 19.4), thereby effectively mitigating the "long-form hallucination" issue exacerbated by language bias.

## D.5. Detailed Analysis of LBP's Impact on General Capabilities

To provide a more comprehensive assessment of LBP's impact on linguistic fluency and response quality, we present two additional sets of experimental data.

First, we provide supplementary analysis using the *Informativeness* metric from MMHalBench. The MMHalBench score (ranging from 0 to 6) is derived from two factors: hallucination status (scores 0–2 indicate hallucination, while 3–6 indicate valid responses) and information richness (higher scores reflect greater alignment with the ground truth in terms of detail). *Informativeness* serves as a specific metric to quantify the fluency and richness of the model's output. Formally, it is

*Table 15.* Stratified analysis of hallucination metrics on the AMBER Generative Task across different output lengths. LBP demonstrates robust performance improvements, particularly in longer context windows.

| Length | DPO | | | | LBP (Ours) | | | |
|---|---|---|---|---|---|---|---|---|
| | CHAIR ↓ | Cover ↑ | Hal ↓ | Cog ↓ | CHAIR ↓ | Cover ↑ | Hal ↓ | Cog ↓ |
| 16 | 2.0 | 23.5 | 2.8 | 0.3 | **1.7** | 23.5 | **2.5** | 0.3 |
| 32 | 2.4 | 35.4 | 6.8 | 0.5 | 2.4 | **35.9** | **6.6** | 0.5 |
| 64 | 3.1 | **44.0** | 12.8 | 0.9 | **2.7** | 43.1 | **11.7** | **0.6** |
| 128 | 4.3 | **55.3** | 24.6 | 1.7 | **3.3** | 55.1 | **19.4** | **1.6** |
| 256 | 4.6 | **56.8** | 26.2 | 1.9 | **3.5** | 55.9 | **20.7** | **1.7** |

*Table 16.* Comparison of Informativeness on MMHalBench. LBP achieves the best balance of Score, Hallucination Rate, and Informativeness.

| Model | Method | Score ↑ | HalRate ↓ | Info. ↑ |
|---|---|---|---|---|
| | V-DPO | 2.16 | 0.56 | 0.28 |
| | MFPO | 2.69 | 0.49 | 0.39 |
| LLaVA-v1.5-7B | DPO | 2.11 | 0.66 | 0.36 |
| | DPO$_M$ | 2.42 | 0.54 | 0.35 |
| | **LBP (Ours)** | **2.91** | **0.43** | **0.40** |
| | MFPO | 2.94 | **0.42** | 0.40 |
| LLaVA-v1.5-13B | DPO | 2.61 | 0.53 | 0.40 |
| | DPO$_M$ | 2.83 | 0.46 | 0.40 |
| | **LBP (Ours)** | **3.01** | **0.42** | **0.42** |

*Table 17.* Comparison of average output length and total hallucination count on human evaluation samples.

| Model | Avg. Length | # Hal ↓ |
|---|---|---|
| LLaVA-v1.5-7B | 114.52 | 155 |
| **LBR (Ours)** | **118.32** | **121** |
| DPO | 116.77 | 137 |
| **LBP (Ours)** | **117.36** | **83** |

calculated as:

$$\text{Informativeness} = \frac{\text{Score}}{3} - (1 - \text{HalRate}) \tag{14}$$

Due to space constraints, these results were omitted from the main text. Table 16 presents the combined results for both the baselines and our ablation settings. Second, we report the average output length of the models on the 100 samples used for our human evaluation, as shown in Table 17.

The results in these tables demonstrate that LBP achieves not only the lowest Hallucination Rate but also the highest Informativeness score. This indicates that LBP does not compromise—and in fact potentially enhances—the model's general capability and linguistic fluency. Furthermore, the human evaluation data confirms that our method produces longer average responses with significantly fewer hallucinations compared to the baseline. These findings collectively demonstrate that LBP effectively improves trustworthiness without exerting a negative impact on general capabilities.

### D.6. Impact of the Visual Encoder on Language Bias

We investigated the visual encoder's role by freezing it during LLaVA-NEXT training. While this notably degraded general performance (Table 18), validating that unfreezing the encoder is key to LLaVA-NEXT's success, it surprisingly had negligible impact on the training dynamics of language bias. This suggests that language bias stems primarily from the conditional probability training objective rather than the learnability (plasticity) of the visual encoder itself.

Conversely, encoder architecture plays a significant role. As analyzed in Section D.1, Qwen2.5-VL exhibits a naturally lower tendency for language bias compared to LLaVA-v1.5, likely due to its redesigned, more powerful Vision Transformer. Although proprietary constraints prevent direct LBR testing on Qwen, the demonstrated effectiveness of LBP (Table 14) confirms that residual bias exists in these advanced models. Thus, we infer that LBR would likely remain effective in further optimizing Qwen's instruction tuning.

### D.7. Comparison with Training-Free Baselines

To further evaluate the effectiveness of our approach, we extend our evaluation to compare against recent state-of-the-art training-free baselines designed to mitigate hallucinations in Large Vision-Language Models (LVLMs). These include VCD (Leng et al., 2024), OPERA (Huang et al., 2024b), and VISTA (Li et al., 2025).

As demonstrated in Table 19, our proposed LBP significantly outperforms all evaluated training-free methods across multiple

*Table 18.* Ablation study on the effect of freezing the vision encoder on LLaVA-NEXT-3B. Unfreezing the vision encoder generally yields superior performance across most benchmarks.

| Model | VQA$^{\text{Chart}}$ | VQA$^{\text{Text}}$ | GQA | SQA$^{\text{I}}$ | MME | MMB$_{en}$ | MMMU | MM-Star | CocoCap |
|---|---|---|---|---|---|---|---|---|---|
| LLaVA-NEXT-3B | **21.1** | **56.1** | 61.9 | **71.1** | 1420 | **69.2** | 39.6 | **42.7** | **109.4** |
| w/ Freeze Vision | 20.0 | 54.5 | 61.9 | 70.4 | **1421** | 68.2 | 39.6 | 41.8 | 108.7 |

benchmarks. Notably, LBP achieves the highest MMHalBench Score and the lowest hallucination rates on both Obj HalBench and CHAIR metrics. This substantial performance margin underscores the advantage of explicitly penalizing language bias during the training phase, which inherently recalibrates the model's cross-modal alignment more effectively than purely inference-time interventions.

*Table 19.* Comparison with state-of-the-art training-free baselines. Our proposed LBP consistently achieves superior performance, indicating the effectiveness of training-time bias mitigation.

| Method | MMHalBench | Obj HalBench | CHAIR | |
|---|---|---|---|---|
| | Score ↑ | Hal ↓ | CHAIRs ↓ | CHAIRi ↓ |
| VCD (Leng et al., 2024) | 2.14 | 0.65 | 48.4 | 23.9 |
| OPERA (Huang et al., 2024b) | 2.19 | 0.62 | 45.6 | 22.8 |
| VISTA (Li et al., 2025) | 2.55 | 0.49 | 26.7 | 12.2 |
| **LBP (Ours)** | **2.91** | **0.43** | **12.3** | **6.3** |

## D.8. Disentangling Objective Bias from Data Bias

A critical aspect of our analysis is ensuring that the observed language bias stems from an inherent modality misalignment within the model architecture, rather than from dataset-specific artifacts. We validate this through both the intrinsic diversity of our training sets and empirical validation via data augmentation.

**Dataset Diversity.** Our training sets are highly diverse and representative, mitigating the risk of distribution-specific bias. Specifically, the LLaVA-v1.5-mix665k dataset is meticulously curated from 11 high-quality sources, encompassing academic VQA, OCR, region-level tasks, and multi-turn conversations. This rigorous, wide-ranging distribution ensures the model learns robust representations across varied contexts. Furthermore, in the VLFeedback dataset, the responses originate from 12 distinct VLM architectures (with GPT-4V acting as a judge). This diverse sourcing confirms that language bias is a ubiquitous, multi-model phenomenon rather than a single-model artifact. Finally, RLHF-V provides 5.7k fully human-annotated preference pairs, grounding our findings strictly in human-verified ground truth.

**Human-Curated Data Augmentation.** To further rule out the possibility of data-specific artifacts, we augmented our supervised fine-tuning (SFT) mixture with an additional 160k human-curated captions from the PixMo dataset (Deitke et al., 2025). We compare the baseline model fine-tuned on this augmented mixture (PixMo-SFT) with our regularized model (PixMo-LBR).

As demonstrated in Table 20 and Table 21, PixMo-LBR strictly outperforms the highly capable PixMo-SFT baseline across both general capabilities and hallucination metrics. This consistent performance gain rigorously validates the generalizability of LBR and confirms the universality of language bias, proving that our regularization mechanism remains highly effective regardless of the underlying high-quality data distribution.

*Table 20.* Performance on representative general benchmarks after augmenting the training mixture with the PixMo dataset. PixMo-LBR consistently improves upon the PixMo-SFT baseline.

| Method | MME | MMBench | GQA | TextVQA | MMMU | VisWiz | COCOCap | InfoVQA |
|---|---|---|---|---|---|---|---|---|
| LLaVA 1.5 7B | 1490 | 64.9 | 62.0 | 45.8 | 35.7 | 50.1 | 110.6 | 21.5 |
| PixMo-SFT (Base) | 1501 | 65.2 | 62.0 | 48.7 | 35.2 | 57.6 | 109.3 | 26.2 |
| **PixMo-LBR (Ours)** | **1529** | **65.8** | **62.8** | **49.6** | **36.2** | **59.9** | **111.1** | **28.3** |

*Table 21.* Performance on hallucination benchmarks after PixMo augmentation. LBR effectively reduces hallucination rates across diverse evaluation frameworks.

| Method | MMHalBench | AMBER | | Object HalBench | | |
|---|---|---|---|---|---|---|
| | Score ↑ | Hal Rate ↓ | CHAIRs ↓ | Hal ↓ | CHAIRs ↓ | CHAIRi ↓ |
| LLaVA 1.5 7B | 2.07 | 0.59 | 8.5 | 39.1 | 54.7 | 27.6 |
| PixMo-SFT (Base) | 2.23 | 0.52 | 8.1 | 38.4 | 52.0 | 26.3 |
| **PixMo-LBR (Ours)** | **2.36** | **0.48** | **7.4** | **36.2** | **49.9** | **24.4** |

**Prompt:** Describe the details in this photo.

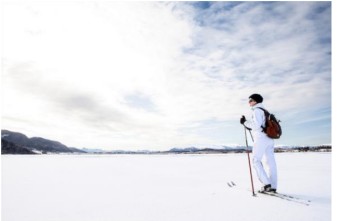

**Model output**

The image is a photograph of a person walking on a frozen lake with a backpack on their back. The person is wearing white clothing and has a red backpack on their back. The sky is blue and there are clouds in the background. The lake is surrounded by mountains and trees. The photograph is taken in the winter and the snow is frozen. The person is walking on the ice with skis and poles. The photograph is taken with a wide angle lens and has a shallow depth of field. The image is taken in the morning and the light is soft and diffused. The photograph is taken with a digital camera and has a high resolution.

*Figure 10.* An example of model output after the Pre-Training (PT) stage. At this point, the model's generations consist of a series of short, descriptive phrases and exhibit negligible language bias.

## E. More Case Studies

### E.1. Model Behavior After Pre-Training

As mentioned in the main text, our analysis focuses on the Visual Instruction Tuning (VIT) stage rather than the initial Pre-Training (PT) stage. Figure 10 provides the justification for this focus. It illustrates that after the PT stage, the model's capabilities are limited to generating short, descriptive captions. Critically, at this stage, the model exhibits only minimal **language bias**. The emergence and growth of this bias primarily occur during the subsequent VIT process, which is why our work concentrates on analyzing and mitigating its effects during that phase.

### E.2. Qualitative Comparison of LBR and Baseline on Human Evaluation

In this section, we present several case studies from our human evaluation (Section 6.2) to qualitatively demonstrate how our LBR method mitigates language bias compared to the baseline VIT-trained model. The following examples are illustrated in Figure 11.

- **Case 1 (Complex Scene):** The baseline model's description of the first image contains multiple hallucinations. It incorrectly claims a man and a baby are **holding** cell phones (two *Action* hallucinations), states there are **two** laptops and **one** book on the table (two *Number* hallucinations), and describes a non-existent **chair** in the background (one *Existence* hallucination). In contrast, the description from our LBR-trained model contains no hallucinations.

- **Case 2 (Object Identification):** In the second example, the baseline model incorrectly identifies an object in the woman's hand as a **"handbag"**, resulting in an *Existence* hallucination. The LBR-trained model correctly describes the object and does not produce any errors.

- **Case 3 (Object Existence):** For the third image, the baseline model hallucinates two non-existent objects, describing **"paper rolls"** next to the toilet and a **"sink"** in the scene (two *Existence* hallucinations). Again, the LBR-trained model's description is free of these hallucinations.

These side-by-side examples provide a clear, intuitive demonstration of LBR's ability to mitigate language bias, leading to a significant reduction in generated hallucinatory content.

### E.3. Qualitative Comparison of LBP and DPO on MMHalBench

To qualitatively demonstrate the advantages of LBP over a standard DPO baseline, we present a series of case studies from the MMHalBench benchmark. We compare the **LLaVA-v1.5-7B** model fine-tuned with LBP against the same model

**Prompt:** *Please help me describe the image in detail.*

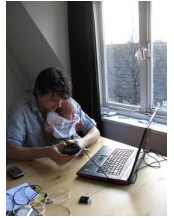

**Baseline**

The image features a man sitting at a dining table with a baby in his arms. The man is holding a cell phone, possibly showing the baby something on the screen. The baby is also holding a cell phone, which is placed in front of them. There are two laptops on the table, one on the left side and another on the right side. A book is also present on the table, located near the right side of the scene. A chair can be seen in the background, positioned behind the man and the baby

**LBR**

The image features a man sitting at a dining table, holding a baby in his arms. The man is looking at a laptop computer placed on the table in front of him. The baby appears to be smiling, enjoying the moment. There are several books scattered around the table, with one near the laptop and another close to the edge of the table. A cell phone can also be seen on the table, possibly belonging to the man or the baby. The scene captures a heartwarming moment between the man and the baby, as they spend time together in a comfortable setting.

**Prompt:** *Please help me describe the image in detail.*

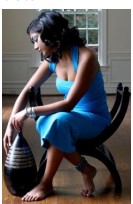

**Baseline**

The image features a woman sitting on a chair, wearing a blue dress. She is holding a brown handbag in her lap, possibly resting her hand on it. The woman appears to be looking down, possibly deep in thought or contemplating something. The chair she is sitting on is positioned in the center of the scene, and the woman is the main focus of the image. The room she is in has a window, which can be seen in the background, providing natural light to the space.

**LBR**

The image features a woman sitting on a chair in a room, wearing a blue dress. She is holding a large, decorative vase in her hands, possibly admiring it or preparing to place it somewhere. The room has a wooden floor, and there is a window nearby, allowing natural light to enter the space. The woman appears to be the main focus of the scene, and her attire and the vase she holds contribute to the overall atmosphere of the room.

**Prompt:** *Please help me describe the image in detail.*

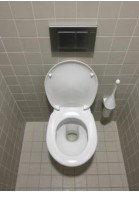

**Baseline**

The image features a white toilet situated in a bathroom stall. The toilet is positioned in the center of the scene, with the lid up. The bathroom appears to be clean and well-maintained. In addition to the toilet, there are two toilet paper rolls placed on the floor, one near the left side of the toilet and the other closer to the right side. The bathroom also has a sink, which is located on the right side of the toilet.

**LBR**

The image features a small, white toilet situated in a bathroom. The toilet is positioned in the corner of the room, with a tile floor surrounding it. The bathroom appears to be clean and well-maintained. In addition to the toilet, there is a toilet brush placed nearby, ready for use. The overall scene gives the impression of a functional and tidy bathroom.

*Figure 11.* Qualitative case studies from our human evaluation, showcasing the benefits of LBR. The figure compares image descriptions generated by the LLaVA-v1.5-7B model trained with LBR against a standard VIT baseline. Hallucinated text produced by the model is highlighted in red.

fine-tuned with DPO, both using the RLHF-V dataset. Figure 12 illustrates examples from the four question-answering tasks (Counting, Environment, Attribute, and Adversarial), while Figure 12 presents three examples from the long-form Holistic description task.

As shown in Figure 12, when presented with targeted questions, the DPO-trained model frequently provides incorrect answers or, even when correct, includes extraneous and irrelevant details that are inconsistent with the image. In contrast, the model trained with our LBP method responds accurately and concisely, without generating superfluous or factually incorrect text.

The distinction is also clear in long-form generation, as shown in Figure 12. The DPO-trained model is prone to errors in specific details. Moreover, as its response lengthens, the error rate tends to increase, in some cases leading to the generation of nonsensical text. Conversely, the LBP-aligned model can describe the image accurately and comprehensively, with a significantly lower propensity for hallucination. It maintains a high degree of factual reliability even when generating long responses.

Taken together, these examples provide a clear, intuitive illustration of how LBP enhances the trustworthiness and visual grounding of LVLMs compared to a standard DPO baseline.

**Prompt:** How many zebras are there in the image?

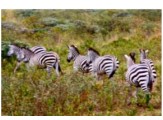

**Baseline**

In the image, there are at least seven zebras, with five of them facing forward and two of them facing backward.

**LBP**

There are six zebras in the image.

**Prompt:** What color are these books?

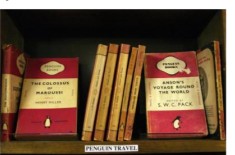

**Baseline**

In the image, there are four books with red, brown, and beige colors, and four books with red, brown, and pink colors. There are also two books with red, brown, and pink colors, and two books with red, brown, and beige colors. In total, there are eight books with red, brown, and pink colors, and four books with red, brown, and beige colors.

**LBP**

The books in the image are red, brown, and white.

**Prompt:** Is this photo taken indoors or outdoors?

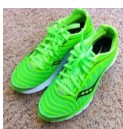

**Baseline**

The photo is taken outdoors, as the carpet in the image is light brown and appears to be a carpeted floor.

**LBP**

The photo is taken indoors, as the carpet appears to be clean and well-maintained.

**Prompt:** Who is sitting on the bench?

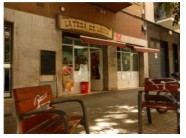

**Baseline**

No one is sitting on the bench in the image, but there are two black metal benches visible in the picture. One of the black metal benches has graffiti on it.

**LBP**

No one is sitting on the bench.

**Prompt:** Describe the details in this photo.

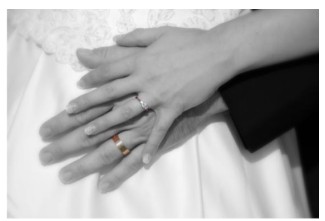

**Baseline**

In the image, there is a close-up view of a person's left and right hands, with the left hand having a ring on the ring finger, and the right hand having a ring on the ring finger as well. The left hand has a ring on the ring finger, and the right hand also has a ring on the ring finger, with the ring on the left hand having a reddish-brown color, while the ring on the right hand is silver and gold. The person's skin is visible, and the image is in black and white, except for the person's skin, which is in color. The person is wearing a black shirt, and the image is in black and white, except for the person's skin, which is in color.

**LBP**

In the image, a man and a woman are standing close to each other, with the woman's hand placed on top of the man's hand. The woman is wearing a wedding ring on her left ring finger, and the man is wearing a wedding ring on his left ring finger as well. The woman is wearing a white dress, and the man is wearing a black shirt.

**Prompt:** Describe the details in this photo.

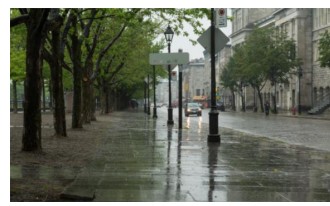

**Baseline**

The image is a black and white photo of a city street on a rainy day, with a sidewalk on the left side of the image. The sidewalk is wet due to the rain, and there is a street on the right side of the image. On the right side of the image, there is a set of stairs with four steps and a red fire hydrant visible in the background. There are trees on the left side of the image, with some of them being leafless.

On the right side of the image, there is a road with a car on it, and the road appears to be wet. There are four street signs on the right side of the image, with two of them located on the right side of the image, and the other two located on the left side of the image.

**LBP**

The image depicts a rainy day in a city, with a wet street reflecting the surrounding environment. The street is lined with trees, creating a pleasant atmosphere despite the rain. There are several cars parked along the side of the street, and a bus can be seen further down the road.

In addition to the vehicles, there are a few pedestrians walking along the sidewalk, some of whom are carrying umbrellas to shield themselves from the rain. A fire hydrant is also visible on the sidewalk, adding to the urban setting. Overall, the scene captures the essence of a typical rainy day in a city, with people going about their daily routines despite the wet conditions.

*Figure 12.* Qualitative examples from MMHalBench comparing our LBP-aligned model against the DPO baseline across a range of short-form (e.g., Counting, Attribute) and long-form (Holistic) tasks. In these examples, correctly identified details are highlighted in green, while hallucinations are highlighted in red.

