# OpenReview forum: "Language Bias in LVLMs: From In-Depth Analysis to Simple and Effective Mitigation"
_ICML.cc/2026/Conference — ICML 2026 regular_

### Official Review · Reviewer_oWV4 · 2026-03-07

**Soundness:** 3
**Presentation:** 3
**Significance:** 3
**Originality:** 3
**Overall Recommendation:** 4
**Confidence:** 4

**Summary:**

To address the severe hallucination issues in Large Vision-Language Models (LVLMs) caused by "language bias", the authors propose two lightweight, data-free training interventions. Based on an analysis of this modality misalignment, they introduce Language Bias Regularization (LBR) for the Visual Instruction Tuning (VIT) stage to constrain the excessive growth of text-only dependence, and a Language Bias Penalty (LBP) for the Direct Preference Optimization (DPO) stage to actively penalize existing language bias. The authors conducted extensive experiments across various LLaVA models to validate the effectiveness of proposed approaches on core hallucination benchmarks (e.g., MMHalBench, AMBER).

**Compliance With Llm Reviewing Policy:**

Affirmed.

**Final Justification:**

Thank you for the authors’ detailed response and additional experiments. The rebuttal addresses all of my previous concerns. I believe the clarification—especially on disentangling objective vs. data bias—would further improve the paper.

**Key Questions For Authors:**

See `Weakness` part.

**Limitations:**

Yes.

**Strengths And Weaknesses:**

## Strengths
1. The paper is generally well-written and easy to follow, with a clear description of the method.
2. This paper conduct comprehensive experiments to validate the effectiveness of proposed method.
3. This paper provides intuitive demonstration cases on visual attention to help understand how the proposed method affects multimodal reasoning and mitigates language bias.

## Weaknesses
1. **Unverified Preservation of Pure-Text Reasoning Capabilities.**

While `Appx.D.7` demonstrates that general LVLM capabilities and informativeness are maintained, there is no evaluation of the models' pure-text reasoning abilities (e.g., MMLU, GSM8K, HumanEval). By penalizing the text-only likelihood $\log \pi_\theta(y|x)$ during visual instruction tuning (LBR), the model is pushed back towards its pre-VIT text distribution. Visual instruction tuning is known to implicitly enhance a model's general instruction-following and logical reasoning capabilities. Forcing a penalty on the text-only prior might cause catastrophic forgetting of complex text-based reasoning. Could the authors include more evaluations on pure-text NLP benchmarks to prove that suppressing visual hallucinations does not come at the cost of degrading the underlying LLM's core logic and reasoning skills.

2. **Lack of End-to-End Pipeline Evaluation for Language Bias Regularization and Language Bias Regularization Synergy.**

The paper logically introduces LBR for the Visual Instruction Tuning (VIT) stage and the Direct Preference Optimization (DPO) stage. However, the experiments evaluate these methods in isolation. It remains unclear whether LBR and LBP can or should be combined into a single, cohesive alignment pipeline.  The authors should provide a sequential training ablation to answer whether the effects of suppressing language bias in the first stage render the second stage penalty synergistic or redundant.

3. **Disentangling Objective Bias from Data Bias.**

The authors attribute the emergence of language bias primarily to the conditional probability MLE/DPO training objectives (i.e., "modality misalignment"). However, `Appx.D.8` shows that freezing the visual encoder barely changes the language bias dynamics, strongly implying that the bias might stem from the *data distribution* rather than just the objective function. If the VIT or DPO datasets contain highly predictable, repetitive text patterns, the model will naturally learn these linguistic shortcuts regardless of the penalty. Could author provide additional discussion on data complexity/augmentation to investigate whether rebalancing the training data (e.g., using responses heavily reliant on fine-grained visual details) naturally mitigates this bias compared to the proposed loss modification.


I'll be more than willing to adjust my evaluation based on authors' responses.

---

> ### Author Rebuttal · Authors · 2026-03-27
>
> We sincerely thank the reviewer for the constructive feedback. Below, we address your concerns with new experiments (full details in final revision). If our response resolves your concerns, we respectfully request you to consider raising your score. We remain available for further discussion.
>
> ---
>
> **Response to W1: Preservation of pure-text reasoning capabilities**
>
> To verify this, Table R1 reports performance on standard pure-text benchmarks. LBR strictly maintains baseline performance, confirming no degradation in text reasoning.
>
> ### **Table R1: Pure-text reasoning capabilities**
>
> |                             | **gsm8k**  | **mmlu**   | **human-eval** |
> | --------------------------- | ---------- | ---------- | -------------- |
> | **LLaVA 1.5 7B (baseline)** | 0.1895     | **0.4954** | **0.1768**     |
> | **LBR**                     | **0.1903** | 0.4936     | 0.1752         |
>
> Theoretically, LBR acts as a gentle regularization on language bias rather than a severe penalty. Furthermore, the joint image-text likelihood continues to optimize normally during training. Together, these mechanisms guarantee that inherent language capabilities are fully preserved.
>
> ---
>
> **Response to W2: End-to-end pipeline evaluation and synergy**
>
> To investigate whether the methods are synergistic or redundant, we conducted a sequential training ablation: applying LBR during SFT, followed by LBP during DPO.
>
> ### **Table R2: Sequential training ablation**
>
> |                    | MMHalBench |          | AMER    |          | Object HalBench |         |
> | ------------------ | ---------- | -------- | ------- | -------- | --------------- | ------- |
> |                    | Score      | HalRate  | CHAIRs  | HalRate  | CHAIRs          | CHAIRi  |
> | **LBP (Baseline)** | 2.91       | 0.43     | 3.5     | 18.5     | 12.3            | 6.3     |
> | **LBR + LBP**      | **2.98**   | **0.40** | **3.2** | **17.9** | **12.1**        | **6.0** |
>
> Table R2 clearly demonstrates that LBR and LBP are highly synergistic. Suppressing language bias in the first stage (LBR) creates a stronger foundation for preference alignment, allowing the combined end-to-end pipeline to further reduce hallucinations and achieve peak performance.
>
> ---
>
> **Response to W3: Disentangling objective bias from data bias**
>
> We clarify that language bias stems from inherent modality misalignment, not dataset-specific artifacts.
>
> **1. Dataset Diversity:**
>
> Our training sets are highly diverse and representative:
>
> - **LLaVA-v1.5-mix665k:** Meticulously curated from 11 **high-quality** sources, covering **Academic VQA**, **OCR**, **Region-level tasks**, and **Conversations**. This rigorous, wide-ranging distribution ensures the model learns robust representations across varied contexts.
> - **VLFeedback:** GPT-4V acts **only as a judge**; the actual responses originate from **12 distinct VLM architectures**. This confirms that language bias is a ubiquitous, multi-model phenomenon, not a single-model artifact.
> - **RLHF-V:** Provides 5.7k **fully human-annotated** preference pairs, grounding our findings in human-verified ground truth.
>
> **2. Human-Curated Data Augmentation:**
>
> To further rule out data-specific artifacts, we augmented our SFT mixture with an additional 160k **human-curated** captions from **PixMo** [1] (AskModelAnything). As shown, PixMo-LBR strictly outperforms the PixMo-SFT baseline in general capabilities (Table R3) and hallucination metrics (Table R4), rigorously validating LBR's generalization and the universality of language bias.
>
> ### **Table R3: Performance on representative general benchmarks**
>
> |                      | **MME**  | **MMBench** | **GQA**  | **TextVQA** | **MMMU** | **VisWiz** | **COCOCap** | **InfoVQA** |
> | -------------------- | -------- | ----------- | -------- | ----------- | -------- | ---------- | ----------- | ----------- |
> | LLaVA 1.5 7B         | 1490     | 64.9        | 62.0     | 45.8        | 35.7     | 50.1       | 110.6       | 21.5        |
> | PixMo-SFT (Base)     | 1501     | 65.2        | 62.0     | 48.7        | 35.2     | 57.6       | 109.3       | 26.2        |
> | **PixMo-LBR (Ours)** | **1529** | **65.8**    | **62.8** | **49.6**    | **36.2** | **59.9**   | **111.1**   | **28.3**    |
>
> ### **Table R4: Performance on hallucination benchmarks**
>
> |                      | MMHalBench |          | AMBER   |          | Object HalBench |          |
> | -------------------- | ---------- | -------- | ------- | -------- | --------------- | -------- |
> |                      | Score      | HalRate  | CHAIRs  | HalRate  | CHAIRs          | CHAIRi   |
> | llava 1.5 7B         | 2.07       | 0.59     | 8.5     | 39.1     | 54.7            | 27.6     |
> | pixmo-SFT (baseline) | 2.23       | 0.52     | 8.1     | 38.4     | 52.0            | 26.3     |
> | pixmo-LBR (ours)     | **2.36**   | **0.48** | **7.4** | **36.2** | **49.9**        | **24.4** |
>
> [1] Molmo and PixMo: Open Weights and Open Data for State-of-the-Art Vision-Language Models, CVPR 25

---

> > ### Author Rebuttal · Reviewer_oWV4 · 2026-04-03
> >
> > Thank you for the authors’ detailed response and additional experiments. The rebuttal addresses all of my previous concerns. I believe the clarification—especially on disentangling objective vs. data bias—would further improve the paper.

---

> > > ### Author Response · Authors · 2026-04-05
> > >
> > > We sincerely thank you for the positive feedback and for dedicating the time to carefully review our rebuttal. We are delighted that our detailed response and the additional experiments have fully resolved your concerns.
> > >
> > > We completely agree with your insightful observation that the clarification—specifically the disentanglement of objective bias from data bias—adds valuable depth to our work. Per your suggestion, we will ensure that these discussions, along with the new empirical results, are prominently incorporated into the final revision of the paper.
> > >
> > > We deeply appreciate your constructive comments and your continued support, which have been instrumental in improving the overall quality of our manuscript.

---

### Official Review · Reviewer_8RJd · 2026-03-12

**Soundness:** 3
**Presentation:** 3
**Significance:** 2
**Originality:** 2
**Overall Recommendation:** 3
**Confidence:** 4

**Summary:**

To address the issue where LVLMs tend to over-rely on textual data while neglecting visual inputs, this work proposes two strategies: LBR, which mitigates bias during instruction tuning, and LBP, which penalizes biased outputs during the DPO process.

**Compliance With Llm Reviewing Policy:**

Affirmed.

**Final Justification:**

This manuscript addresses language bias in multimodal alignment by introducing a language bias regularization term during SFT and DPO. While the approach is straightforward and yields moderate gains, the theoretical framework lacks depth. Though the rebuttal successfully addressed my primary empirical concerns, it introduced further ambiguity: the authors claim to penalize 'guessing' rather than 'language,' yet this distinction remains ill-defined. Furthermore, the authors describe language bias as a 'multimodal routing failure,' which implies an MoE architecture; however, the LLaVA-series models used in this study are not MoE-based.
Considering these theoretical insufficiencies and the limited originality, I maintain the Weak Reject recommendation.

**Key Questions For Authors:**

Please see the weaknesses.

**Limitations:**

Yes

**Strengths And Weaknesses:**

### Strength
* The paper goes beyond mere empirical observation by providing a rigorous quantitative analysis, identifying the root cause of language bias as "modality mismatch" during the training process.
* The proposed methods are straightforward to follow and implement. Their effectiveness has been validated across multiple model architectures.
### Weakness
* The technical novelty appears somewhat constrained. LBR and LBP are primarily formulated as additional regularization terms within existing ViT and DPO objective functions, which may limit the overall contribution.
* An incorrect regularization direction could lead to gradient conflicts during training. This is particularly concerning given LBR’s extreme sensitivity to the hyperparameter $\alpha$; a biased reference model may further narrow the equilibrium space between maintaining general capabilities and reducing hallucinations.

---

> ### Author Rebuttal · Authors · 2026-03-27
>
> We sincerely thank the reviewer for the constructive feedback. Below, we clarify your concerns and provide new experiments (representative results shown due to space limits; full details in final revision). If this resolves your concerns, we respectfully request raising your score (currently 3: Weak Reject). We remain available for further discussion.
>
> > **Comment 1**: The technical novelty appears somewhat constrained. LBR and LBP are primarily formulated as additional regularization terms within existing ViT and DPO objective functions, which may limit the overall contribution.
>
> **Response**:  We respectfully argue that simplicity is a major strength, not a limitation. Achieving significant gains without complex architectural overhead offers three key advantages:
>
> 1. **Insight-Driven:** Our core contribution is identifying the root cause of long-form hallucinations (over-reliance on language priors). LBR and LBP are elegant, targeted solutions to this exact problem.
> 2. **Superior Generalization:** As **Reviewer oWV4** noted, this straightforward formulation **avoids overfitting** and drives our exceptional performance across diverse benchmarks.
> 3. **Zero Inference Overhead:** As pure training-time regularizers, LBR and LBP suppress hallucinations with **zero added latency** during deployment.
>
> Ultimately, providing a highly effective, **plug-and-play** solution to a fundamental problem represents a substantial contribution to the community.
>
> ---
>
> > **Comment 2**: An incorrect regularization direction could lead to gradient conflicts during training. This is particularly concerning given LBR’s extreme sensitivity to the hyperparameter ;
>
> **Response**:  We thank the reviewer for this insightful point. We clarify two key aspects regarding LBR's sensitivity and potential gradient conflicts:
>
> **1. Robustness across a wide range of $\alpha$:**
>
> LBR's perceived "extreme sensitivity" is only relative to LBP; in absolute terms, it remains highly robust. As shown in Tables R1 and R2, across a broad four-order-of-magnitude range ($\alpha \in [10^{-7}, 10^{-3}]$), LBR consistently outperforms the SFT baseline on most benchmarks. This confirms the method is not brittle and effectively resists severe gradient conflicts even under fixed-weight settings.
>
> ### **Table R1: Ablation on $\alpha$ across representative benchmarks**
>
> |               | **MME**  | **MMBench** | **GQA**  | **TextVQA** | **SQA**  | **MMMU** | **VisWiz** | **COCOCap** |
> | ------------- | -------- | ----------- | -------- | ----------- | -------- | -------- | ---------- | ----------- |
> | Baseline      | 1490     | 64.9        | 62.0     | 45.8        | 66.8     | 35.7     | 50.1       | 110.6       |
> | $\alpha=1e-3$ | 1492     | 64.8        | 62.3     | 45.7        | 69.9     | 36.3     | **55.2**   | 110.8       |
> | $\alpha=1e-4$ | 1517     | 64.9        | 62.4     | 45.6        | 68.3     | 36.5     | 52.4       | **116.3**   |
> | $\alpha=1e-5$ | **1525** | **65.3**    | **62.7** | **46.0**    | 69.4     | **37.2** | 54.0       | 112.1       |
> | $\alpha=1e-6$ | 1484     | 65.0        | 62.6     | 45.6        | **70.7** | 35.9     | 54.6       | 110.1       |
> | $\alpha=1e-7$ | 1457     | 64.8        | 62.5     | **46.0**    | 68.9     | 36.2     | 53.1       | 112.2       |
>
> **2. Eliminating gradient conflicts via Dynamic Scheduling:**
>
> While fixed weights are robust, we actively investigated your specific concern. Our dynamics analysis (Appendix D.3) shows that bias mitigation requires strong early suppression, but maintaining this intensity can cause late-stage over-constraint.
>
> To explicitly avert gradient conflicts, **Appendix D.5** introduces a **Cosine Annealing** schedule (decaying $\alpha$ from $10^{-4}$ to $10^{-6}$). As shown in Tables 13 and 14, this dynamic approach perfectly balances regularization, avoids optimization conflicts, and yields superior performance over the fixed-weight baseline.
>
> ---
>
> > **Comment 3**: a biased reference model may further narrow the equilibrium space between maintaining general capabilities and reducing hallucinations.
>
> **Response**: We respectfully clarify a fundamental **misunderstanding** regarding our **reference models**. Specifically, the reference is simply the base pre-trained model for LBR, and the standard SFT model (e.g., LLaVA 1.5) for LBP. We completely agree these models inherently possess language bias—in fact, identifying and quantifying this exact bias is the **core motivation** of our paper.
>
> Crucially, our methods do not passively *inherit* this bias. Instead, LBR and LBP utilize these reference models to explicitly isolate and *penalize* text-only language priors. Therefore, the reference bias does not **"narrow the equilibrium space"** between general capabilities and hallucination reduction. Rather, it serves as the precise mathematical target our objective functions successfully neutralize, ultimately *expanding* the model's capacity for faithful generation.

---

> > ### Author Rebuttal · Reviewer_8RJd · 2026-04-03
> >
> > Why regularizing for language bias does not degrade linguistic proficiency, and may even enhance performance on text-intensive tasks？

---

> > > ### Author Response · Authors · 2026-04-05
> > >
> > > We sincerely thank the reviewer for this insightful follow-up. Below, we clarify the theoretical distinction between "language bias" and "linguistic proficiency," detailing why our method inherently preserves the latter while substantially boosting performance on text-intensive visual tasks:
> > >
> > > **1. Theoretical Boundary: Penalizing "Guessing," Not "Language"**
> > >
> > > Linguistic proficiency stems from the LLM's inherent syntactic and reasoning weights. "Language bias," in contrast, represents a **multimodal routing failure**—the model relies solely on text priors to guess the next token instead of attending to visual features. Crucially, our bias mitigation methods (LBR and LBP) operate in **direct synergy** with standard SFT and DPO objectives. While our methods act to suppress or penalize the text-only prior $\pi(y|x)$, the primary training objectives simultaneously maximize the joint multimodal likelihood $\pi(y|x, v)$. This synergistic dynamic—**penalizing pure-text guessing while rewarding multimodal accuracy**—does not alter the LLM's intrinsic reasoning pathways. Instead, it explicitly forces the model to shift its attention toward the visual inputs ($v$) to minimize the overall loss, thereby ensuring tight visual grounding before speaking.
> > >
> > > **2. Empirical Proof of Preserved Proficiency**
> > >
> > > To definitively confirm that fundamental language capabilities remain intact, we conducted new pure-text evaluations targeting fluency (WikiText), deep reasoning (Winogrande), and contextual reading comprehension (SQuAD-v2).
> > >
> > > ### **Table R2: Pure-text proficiency evaluation**
> > >
> > > |                         | **WikiText (PPL) ↓** | **Winogrande ↑** | **SQuAD-v2 ↑** |
> > > | ----------------------- | -------------------- | ---------------- | -------------- |
> > > | **LLaVA 1.5 7B (Base)** | 11.86                | 70.72            | **18.10**          |
> > > | **LBR (Ours)**          | **11.73**            | **70.86**        | 17.94          |
> > >
> > > As shown in Table R2, LBR maintains (and even slightly improves) performance across dimensions. This confirms that our regularization strictly suppresses shallow statistical guessing with **zero degradation to core linguistic proficiency**.
> > >
> > > **3. Enhancing Text-Intensive Tasks: Combating the "Snowball Effect"**
> > >
> > > Text-intensive visual generation suffers heavily from a **"snowball effect."** As the generated text lengthens, standard VLMs rely increasingly on their own generated linguistic context (language priors) while visual attention decays, causing severe hallucinations. To empirically prove this, we partitioned the generative hallucination metrics by text length:
> > >
> > > ### **Table R3: Hallucination metrics across varying generation lengths**
> > >
> > > |            |             | DPO (Baseline) |             | LBP (Ours) |
> > > | ---------- | ----------- | -------------- | ----------- | ---------- |
> > > | **Length** | **CHAIR ↓** | **Hal ↓**      | **CHAIR ↓** | **Hal ↓**  |
> > > | **16**     | 2.0         | 2.8            | **1.7**     | **2.5**    |
> > > | **32**     | 2.4         | 6.8            | **2.4**     | **6.6**    |
> > > | **64**     | 3.1         | 12.8           | **2.7**     | **11.7**   |
> > > | **128**    | 4.3         | 24.6           | **3.3**     | **19.4**   |
> > > | **256**    | 4.6         | 26.2           | **3.5**     | **20.7**   |
> > >
> > > As shown in Table R3, the baseline's hallucination rate **escalates dramatically at longer lengths** (Hal reaches 26.2 at length 256). By actively penalizing language priors, LBP enforces continuous visual grounding, **significantly suppressing long-form hallucinations** (Hal drops to 20.7). Consequently, mitigating language bias prevents the model from being "hijacked" by its own autoregressive inertia, naturally boosting performance in extended multimodal tasks.
> > >
> > > **4. Encouraging Visual Grounding via Logit Calibration in Text-Rich Images**
> > >
> > > Beyond mitigating the snowball effect in long-form generation, our method also helps improve performance on tasks like TextVQA and OCR through logit calibration.
> > >
> > > In text-intensive images, ground-truth answers often consist of **irregular, non-semantic strings** (e.g., alphanumeric codes or obscure brand names). In the LLM's pure-language prior distribution, these less common tokens naturally possess lower probabilities. Consequently, a baseline VLM influenced by language priors may **favor the logits of common, semantically plausible words**. This can lead the model to "auto-complete" a guess based on context rather than accurately reading the actual image.
> > >
> > > By regularizing these language priors, our method helps balance the probability distribution. This mitigates the tendency of the LLM's vocabulary habits to overshadow the softmax output, **encouraging the model to rely more on visual evidence** to elevate the logits of the correct tokens. Ultimately, moderating language bias serves as a helpful visual-grounding mechanism, facilitating the model's shift from relying on semantic guesses to focusing on the visual text.

---

### Official Review · Reviewer_13V5 · 2026-03-17

**Soundness:** 3
**Presentation:** 3
**Significance:** 3
**Originality:** 2
**Overall Recommendation:** 4
**Confidence:** 4

**Summary:**

In this work, the authors explore language bias in VLMs – that is, the tendency of VLMs to ignore visual input during generation, resulting in failures like hallucination.  Prior work has shown evidence for this bias in the form of minimal attention to image tokens during response generation.  Here, the authors argue that this bias is caused by modality misalignment introduced during training – they claim that visual instruction tuning and direct preference optimization prioritize textual reconstruction, favoring language modeling over multimodal understanding.  They present two measures of this phenomenon that they call reward (improvement in a model in the image conditioned setting tasks) and language bias (improvement in a model in the image unconditioned setting for image tasks) and show that the gap between the two is minimal for instruction tuning and inverted for DPO.  Then, the authors show that by directly regularizing these measures during post-training (by penalizing the magnitude of the language bias term), they can reduce hallucination rates in their models without compromising task performance.

**Compliance With Llm Reviewing Policy:**

Affirmed.

**Final Justification:**

The authors' rebuttal thoroughly engaged with my review, expanding their baseline coverage, demonstrating the robustness off their results across multiple seeds and showing that the gains yielding by their method is not simply an artifact of dataset noise.  As such, I feel confident in raising my scores for both the soundness and the significance of the author's contribution.

As the authors are more or less optimizing a measure previously employed for training free guidance, my assessment of the originality of the contribution remains unchanged.

Still, the reductions in hallucinations the authors demonstrate are robust and significant so my overall recommendation (to 4: weak accept).

**Key Questions For Authors:**

Could the authors speak to / address the weaknesses in soundness and significance above please?

**Limitations:**

Yes

**Strengths And Weaknesses:**

*Soundness*

S1.  I like the gap measure the authors introduce for measuring over-reliance on text during image-conditioned generation.  It's reasonably principled, easy to understand and allows detecting a downstream phenomenon during training.

S2.  The authors select a fairly comprehensive set of benchmarks for general purpose image understanding and hallucination detection.

S3.  I was also happy to see ablations around the hyperparameters the authors introduce in their regularizer.

S4.  The reductions in hallucination due to BLP seem real and the human study is a welcome addition (especially the investigation into the kinds of error types introduced by DPO).

W1.  Inadequate baselines: while the authors reference some training free methods for reducing language bias, they do not actually compare their method against these.  While such methods might be computationally more intensive at inference time, they are certainly easier to apply to large, out-of-the-box models.  Moreover, many of these methods employ similar ratios (e.g., image conditioned vs image unconditioned) to the ones that authors attempt to optimize directly here.  Thus, understanding the strengths / weaknesses of the author's proposed methods requires their inclusion.

W2.  Overly broad claims: the gains the authors show on standard image understanding benchmarks (Table 1) are extremely modest and could very well be statistical noise.  Are the authors able to do multiple training runs and report means and variances?  Otherwise, the authors should consider narrowing claims re: efficacy of their method to something like "reduces hallucination and does not degrade image understanding"

W3.  Potential confound due to reliance on synthetic data: I believe that the texts in both the instruction tuning dataset (LLaVA) and the preference dataset (VLFeedback) the authors use for their experiments are GPT-generated (which likely exhibits the same language bias).  In fact, LLaVA was GPT generated without any image conditioning (they condition on captions and bounding box annotations).  Thus, the language bias the authors measure during training may not be intrinsic to the training setup but may instead be an artifact of the data used.  Can the authors speak to this at all?  An interesting and compelling addition could be to use the captioning data released as part of the Pixmo dataset as a high quality, human produced alternative to better isolate what drives the measured language bias.

W4.  (minor) While the human study on detailed image description is a welcome addition in terms of error analysis, it would be strengthened with the inclusion of evaluation of detailed image description benchmarks for which automated evaluation metrics exist.  This would improve the replicability of the paper's results.

*Presentation*

S1.  The writing and figures are clear and easy to follow.

*Significance* / *Originality*

W1.  Several works (including the ones below which are currently missing from the paper) have investigated debiasing VLMs through the use of text only priors or noisy image priors.  While these methods certainly need to be included as training free baselines (see Soundness, W1 above), the authors need to clarify the novelty of their analysis/method compared to this work:

- Revisiting the Role of Language Priors in Vision-Language Models (ICML 2024): https://arxiv.org/abs/2306.01879
- Counterfactual VQA: A Cause-Effect Look at Language Bias (CVPR 2021): https://openaccess.thecvf.com/content/CVPR2021/html/Niu_Counterfactual_VQA_A_Cause-Effect_Look_at_Language_Bias_CVPR_2021_paper.html
- Debiased Visual Question Answering from Feature and Sample Perspectives (Neurips 2021): https://proceedings.neurips.cc/paper/2021/hash/1f4477bad7af3616c1f933a02bfabe4e-Abstract.html

---

> ### Author Rebuttal · Authors · 2026-03-27
>
> We sincerely thank the reviewer for the constructive feedback. Below, we clarify your concerns and provide new experiments (representative results shown due to space limits; full details in final revision). If this resolves your concerns, we respectfully request raising your score (currently 3: Weak Reject). We remain available for further discussion.
>
> ---
>
> **Response to W1: Comparison with training-free baselines**
>
> We initially omitted training-free methods due to their marginal gains. Per your suggestion, Table R1 shows our LBP significantly outperforms recent SOTA baselines across all metrics.
>
> ### **Table R1: Comparison with training-free baselines.**
>
> | **Method**     | MMHalBench |          | Obj HalBench |         |
> | -------------- | ---------- | -------- | ------------ | ------- |
> |                | Score      | Hal      | CHAIRs       | CHAIRi  |
> | VCD [1]        | 2.14       | 0.65     | 48.4         | 23.9    |
> | OPERA [2]      | 2.19       | 0.62     | 45.6         | 22.8    |
> | VISTA [3]      | 2.55       | 0.49     | 26.7         | 12.2    |
> | **LBP (Ours)** | **2.91**   | **0.43** | **12.3**     | **6.3** |
>
> [1] Mitigating Object Hallucinations in Large Vision-Language Models through Visual Contrastive Decoding, CVPR 24
>
> [2] OPERA: Alleviating hallucination in multimodal large language models via over-trust penalty and retrospection-allocation, CVPR 24
>
> [3] The Hidden Life of Tokens: Reducing Hallucination of Large Vision-Language Models via Visual Information Steering, ICML 25
>
> ---
>
> **Response to W2: Statistical significance of performance gains**
>
> To rule out statistical noise, Table R2 reports the mean and std of 3 independent SFT runs across 8 representative benchmarks.
>
> ### **Table R2: Stability of LBR across benchmarks**
>
> |                | MME          | MMBench      | GQA          | TextVQA      | SQA          | MMMU         | VisWiz       | COCOCap       |
> | -------------- | ------------ | ------------ | ------------ | ------------ | ------------ | ------------ | ------------ | ------------- |
> | Baseline       | 1490         | 64.9         | 62.0         | 45.8         | 66.8         | 35.7         | 50.1         | 110.6         |
> | **LBR (Ours)** | **1524**±3.5 | **65.3**±0.2 | **62.8**±0.2 | **46.0**±0.1 | **69.2**±0.3 | **37.1**±0.3 | **53.9**±0.4 | **112.0**±0.5 |
>
> These results confirm our gains are **statistically robust, not random noise**. Modest improvements are expected here, as these benchmarks primarily evaluate **short-text** scenarios. Crucially, LBR targets **long-form generation**, where it substantially reduces hallucinations. Additionally, our $\alpha$ ablation (see **Reviewer 8RJd, Comment 2**) confirms consistent gains across broad settings, reinforcing this stability. Thus, these evaluations validate our core claim: LBR stably enhances general understanding while significantly mitigating long-form hallucinations.
>
> ---
>
> **Response to W3: Potential confound from synthetic data**
>
> We respectfully clarify that LLaVA and VLFeedback are **not** solely GPT-generated without image conditioning. Per your suggestion, we incorporated 160k high-quality human Pixmo captions (*AskModelAnything*) into our SFT mixture to evaluate LBR against standard SFT. Due to space limits, please refer to **Reviewer oWV4 (W3)** for **full empirical results and clarifications on dataset complexities**.
>
> ---
>
> **Response to W4: Inclusion of automated evaluation metrics**
>
> We respectfully clarify that Tables 3 and 11 **already** report automated generative benchmarks (MMHalBench, AMBER, Object HalBench). As Section 6.2 explains, because these automated metrics struggle with complex long-form hallucinations, our human study is designed to *complement*—not replace—them for deeper error analysis.
>
> ---
>
> **Response to W1: Novelty compared to the mentioned works**
>
> We thank the reviewer for highlighting these relevant works. Our approach introduces fundamental innovations across three dimensions:
>
> 1. **Paradigm:** The referenced works target discriminative tasks like **multi-class VQA** [CVPR '21, NeurIPS '21] or **retrieval** [ICML '24] using **legacy models**. In contrast, we tackle **modern Generative LVLMs** to mitigate multimodal hallucinations during **long-form generation**.
> 2. **Root Cause:** They treat bias as static dataset artifacts. We conceptually redefine it as a **Modality Misalignment in Training Dynamics**, shifting the focus to the inherent conditional probability paradigm of LVLMs.
> 3. **Intervention:** The mentioned methods rely on complex inference-time hacks [ICML '24, CVPR '21] or heavy architectural modifications [NeurIPS '21]. Instead, our plug-and-play loss functions intervene directly during training. This directly suppresses text-only probability drift while requiring **zero auxiliary models, extra data, or inference overhead**.
>
> **Given these fundamental task and architectural disparities, direct empirical comparisons are infeasible.** We kindly ask for your understanding.

---

> > ### Author Rebuttal · Reviewer_13V5 · 2026-04-03
> >
> > I’d like to thank the authors for their thorough engagement with their review.  Generally speaking, they addressed all of my most pressing concerns — I appreciate the inclusion of training free baselines, the inclusion of standard deviations for their runs, and evaluation of their method using the higher quality Pixmo dataset.  I’ll raise my evaluation of the soundness and significance of their work (to 3) and my overall recommendation (to 4).
> >
> > Regarding my point about automated evaluation metrics, apologies, I should have been more specific.  The authors employ automated evaluation metrics for benchmarks that are focused on shorter texts but state that detecting hallucinations in complex, long-form texts requires human evaluation.  While understandable, metrics for evaluating detailed image descriptions have been introduced in the last 1-2 years (see links below) and evaluating your approach using these metrics would be a welcome improvement to the replicability of your findings.
> >
> > https://arxiv.org/abs/2503.07906
> > https://aclanthology.org/2025.findings-acl.724/
> > https://arxiv.org/abs/2510.19060

---

> > > ### Author Response · Authors · 2026-04-05
> > >
> > > We sincerely thank you for your thorough engagement throughout the review process and for raising your evaluation scores. We are thrilled that our previous additions have effectively addressed your primary concerns.
> > >
> > > Regarding the automated evaluation metrics for long-form generation, thank you for the clarification and for providing these excellent references. You make a fair point: while our human evaluation provides reliable and nuanced insights, complementing it with these newly introduced automated metrics would be a valuable addition to further enhance the replicability of our research.
> > >
> > > We will carefully study the recent papers you highlighted. We will certainly expand our discussion to include these advancements in the final manuscript, and we will strive to incorporate these automated experiments in the camera-ready version if time permits.
> > >
> > > Thank you again for your constructive guidance and your continued support of our work!

---

### Decision · Program_Chairs · 2026-04-30

**Decision:**

Accept (regular)

**Comment:**

This paper presents a systematic investigation of language bias in vision-language models, identifying modality misalignment during training as its root cause. It shows that common training paradigms can tend to favor textual optimization, leading to over-reliance on language and increased hallucination. To address this, lightweight methods are proposed for instruction tuning and for preference optimization. Extensive experiments across multiple models and benchmarks demonstrate the effectiveness of the methods and justify the paper's claims well.

It received review comments from three reviewers. Two reviewers acknowledged that the raised concerns had been addressed. One reviewer acknowledged that most of the concerns were solved and proposed a follow-up question. The AC checked the review comments, all questions, and all author feedback, and found that the current form meets the acceptance line. Therefore, this work is recommended for acceptance. The final version of the work could be further improved based on the reviewers' comments and suggestions.